# An Empirical Correction Model for Remote Sensing Data of Global Horizontal Irradiance in High-Cloudiness-Index Locations

**Martín Muñoz-Salcedo [1], Fernando Peci-López [2,3,*] and Francisco Táboas [2]**

[1]  Facultad de Ciencias e Ingeniería, Universidad Estatal de Milagro, Milagro 091051, Ecuador
[2]  Departamento de Química-Física y Termodinámica Aplicada, Universidad de Córdoba, 14014 Córdoba, Spain
[3]  International Researcher, Universidad Ecotec, Guayaquil 092302, Ecuador
*  Correspondence: fernando.peci@uco.es; Tel.: +34-647915281

**Abstract:** Facing the energy transition, solar energy, whether thermal or electric, is currently one of the most viable alternatives, due to its technological maturity and its ease of operation and maintenance compared to other renewable energies. However, before its implementation, it is necessary to assess its potential. Remote sensing represents one of the low-cost solutions for solar energy assessment. Nevertheless, cloud cover is a main problem when validating the data. This study identifies satellite GHI profiles that cannot be used in energy production simulation. The validation is performed using parametric and non-parametric statistical tests. From the profile identified as invalid for simulation purposes, a site-adaptation methodology is proposed based on statistical learning using the machine learning algorithms "Best subset selection" and "Forward Stepwise Selection". Linear and non-linear heuristic models are also proposed. The final AS7 model is selected through RMSE, MBE and adjusted $R^2$ indicators and is valid for any sky condition. The results show an increase in $R^2$ from 0.607 to 0.876.

**Keywords:** solar resource assessment; cloudiness empirical model; site-adaptation; remote sensing

## 1. Introduction

The current effects of climate change place the world in a difficult and undeniable position. The consequences of the use of fossil fuels and their impact on the ecosystem are leading to a no-return scenario. Due to the increasing trend in the use of fossil fuels in the production of electrical and thermal energy, and the post-pandemic global economic recovery, a significant increase in energy consumption and greenhouse gas emissions is foreseen [1]. Faced with this reality, it is necessary to take timely and, in some cases, emerging actions to mitigate environmental impact. The development of new technologies, a higher penetration of renewable energies, and alternative fuels represent some of these actions. The fight against climate change, following its global impact and disruption of the world's economies, has prompted projects for sustainable development. More and more countries invest in climate-friendly systemic solutions. The commitments made at the last COP26 climate summit [2] indicated actions to gradually reduce carbon emissions by 45% within a decade. Additionally, energy plans are to be drawn up by 2022 instead of 2025.

Therefore, the energy transition towards decarbonization must consider the use of clean energy sources and more efficient technologies, and provide a resilient, independent, secure, and reliable energy supply model in response to climate uncertainty. Greater penetration of non-conventional renewable energies requires that electricity generation projects be located in areas of high energy potential; thus, it is crucial to assess surface renewable resources.

The meteorological variable estimations, such as wind speed and direction, ambient temperature, humidity, atmospheric pressure, and solar irradiance, are the determinations necessary for all phases of a renewable energy project. The planning phase allows the determination of the long-term variability of the resource and forecast energy production through simulation. The operation phase allows the observation of the system performance and the identification of problems. In the absence of weather stations, which require high investment, operation and maintenance costs, there are alternative tools capable of evaluating several meteorological variables.

One of the most critical meteorological variables is solar irradiance. To obtain information regarding solar irradiance, radiometric sensors are necessary, but in their absence, low-cost alternatives such as meteorological yearbooks, solar atlases or data from remote sensors are available. Geostationary remote sensors, such as the GOES satellite [3], capture reflectivity images of the Earth's surface through its different bands, and through the application of a physical solar model, quantify the solar resource present in the satellite coverage. They are currently capable of providing spatio-temporal resolutions of up to 1 km in 30 min profiles. As technological development advances, satellite resolution is becoming higher and prediction models more accurate. However, one of the main issues with solar irradiance estimation is cloud cover [4]. The absorption and scattering effect present in the atmospheric components, attenuates the flux of extraterrestrial radiation from the sun. For this reason, predictive models are used. The input data to these models can be obtained from nearby stations and satellite estimates. According to [5], empirical, semi-empirical, and physical models can be distinguished in a deterministic way. The empirical model uses both, surface measurements and satellite estimates taken long term, to generate a regression model between these two variables. Semi-empirical models use a cloudiness parameter in the visible satellite channels, which is fitted to a global horizontal irradiance (GHI) clear-sky model, so corrections can be added for altitude and atmospheric turbidity. To obtain the irradiance at the Earth's surface, physical models are used to retrieve from satellites the properties of clouds and aerosols through the multiple channels of their bands; the information is calculated in a radiative transfer model.

GHI in minute or hourly profiles is necessary for the forecast simulation of energy production, as computer programs require fine resolution input data. In [6], the authors proposed a probabilistic methodology to obtain the irradiance for day-ahead prediction, using a stochastic differential equation (SDE) model. The usefulness of the model suggests its application in very short time intervals, especially when it comes to energy markets, micro-grids or energy storage dispatch systems, where predictions should be as close to reality as possible.

Nowadays, several studies are focused on improving irradiance from satellite-derived or re-analysis. As mentioned above, aerosols, cloudiness and temporal variability represent the main issues to validate satellite imagery data before its application. Satellite images can quantify the solar resource through models that describe the aerosols present in the atmosphere and the water vapor within the clouds. The modeled irradiance is often inaccurate. For this reason, different techniques called "site-adaptation" have been developed to reduce the biases of satellite-derived.

### 1.1. State of the Art

Site-adaptation may be performed by applying one of three main procedures based on regression methods, quantile mapping or a combination of both. The work carried out in [7–9] details the techniques and methodologies used for site-adaptation. This requires at least one year of high-quality surface measurements. Solar geometry variables such as solar elevation angle or zenith angle are widely used as predictors of the new correction model. In addition, atmospheric variables such as aerosol optical depth [10], Linke turbidity index [11] or air mass could be added. Due to different atmospheric components, terrain and site meteorological conditions, there is no universal model to reduce satellite radiation

errors at a specific location. Therefore, it is necessary to develop studies that evaluate different types of climates and sites.

In [12] an inventory of approaches for site-adaptation of satellite-based DNI and GHI optimize SolarGIS data through the rescaling method and fitting cumulative distribution function. The results show that if the differences are larger (the bias is greater than 4% for the GHI and 7% for the DNI) the use of rescaling or cumulative distribution function methods can introduce strong inconsistencies. The authors in [13] evaluate the accuracy of modeled solar irradiance series, their adaptation to specific sites and their long-term behavior for GHI and DNI. Greater inaccuracy of DNI in relation to GHI of modeled data with respect to surface measurements. The biases found in the initial DNI model (on average 6%, up to 15%) are drastically reduced by applying side-adaptation techniques, up to 15%), both based on linear regression (~2%) and statistical learning (~0%).

The use of Model Output Statistics (MOS) and Kalman filter in [14] to improve GHI modeled by Weather and Research Forecasting (WRF), reduced the systematic model error for all sky conditions, in comparison with surface measurements in Paraguay. The comparison of bias with respect to the raw model provides an improvement for summer from 68 to $-1.1$ W m$^{-2}$ (27% to $-1.5$%) and spring from 60 to 0.8 W m$^{-2}$ (29% to 0.3%), which presents the seasons with the most systematic model errors. RMSE for summer was from 207 to 169 W m$^{-2}$ (71% to 61%) and spring was from 198 to 175 W m$^{-2}$ (75% to 68%).

The technique of Orthogonal Distance Regression (ODR) [15] is used to evaluate GHI and reduce the bias from the tool of re-analysis COSMO-REA6 in ten European regions. The results show strong improvement in MAE from 83.2 to 59.4 W m$^{-2}$ for Sede Boqer and become slightly worse (97.9–105.2 W m$^{-2}$) for Lerwick. Furthermore, in [16] a method for site-adaptation in Colombia through machine learning and solar radiation prediction using deep learning is proposed. As a consequence, a better performance for the database was improved through machine learning techniques over the commonly used statistical methods for site-adaptation. The performance of the models depends on meteorological data, which have different behavior depending on the location.

Pre-processing technique through Cloud Index Method (CIM) and pos-processing through linear regression correction are carried out in [17]. The study evaluates the performance of ESRA and McClear models in locations out of satellite coverage. Site-adaptation of Pampa Húmeda is applied to the Heliostat-4 GHI estimates. The results of the site-adapted McClear model provide the best performance with the lowest rRMSD and KSI at all locations. However, the McClear model in all sky conditions is sensitive to changes in the atmosphere such as cloudiness.

In [18], to validate GHI from the Copernicus Atmosphere Monitoring Service (CAMS), supervised machine learning algorithms are applied to site-adapt the CAMS in Patras, Greece. Techniques such as Fully Connected Neural Networks, Extreme Gradient Boosting machines (XGBoost), Random Forests (RF), Elastic Net regression (GLMNET), Multivariate Adaptive Regression Splines (MARS) and Support Vector Regression (SVR) are used. The results show significant systematic and dispersion errors exist for all sky (MBE = 22.8 W m$^{-2}$, RMSE = 74 W m$^{-2}$) and cloudy (MBE = 48.2 W m$^{-2}$, RMSE = 106.5 W m$^{-2}$) conditions. Site-adaptation reduces MBE and RMSE at various Solar Zenith Angles (SZAs) and Cloud Fraction (CF) cases. The lowest RMSE values are revealed for the tree-based Machine Learning Algorithms (MLAs). The improvement in RMSE extends between 37.1 W m$^{-2}$ and $-9.3$ W m$^{-2}$.

Site-adaptation studies have been carried out in different areas and climates worldwide. In most cases, different methodologies have improved satellite-derived and re-analysis data. The literature highlights deterministic correction models ranging from pre-processing of changes in data distribution and simple linear regressions, to the advanced application of artificial intelligence in probabilistic techniques. The latter have not necessarily had the best results to reduce dispersion or bias. On the other hand, the effects on climate and topography variation present in coastal areas near the Andean region [19] have not yet been evaluated. Thus, future work may be of significant interest to cover these

regions. Therefore, as site adaptation techniques continue to progress and satellite-derived validation achieves greater coverage throughout the world, regional adaptation models could be possible.

### 1.2. Ecuadorian Context as a Case Study

Ecuador is a nation whose state policy has included a change in the energy matrix, and whose current electricity generation is based on thermal and hydroelectric power plants, with low penetration of non-conventional renewable sources [20]. For the assessment of renewable resources, there is a dispersed and weak meteorological station network and access to its information is restricted. On the other hand, alternative information tools include: the Solar Atlas [21], Wind Atlas [22], and Bioenergy Atlas [23], to cite a few examples. Although these tools allow a general visualization of the resource potential available in the Ecuadorian territory, it is not possible to obtain fine resolution profiles. Thus, Geographic Information Systems (GIS) may be an interesting alternative. However, before using data from these sources, it is necessary to validate them.

For many years, the tools available for estimating renewable resources in Ecuador have not been updated in line with the improvements in the resolution of remote sensors. In [24], the methodology used for evaluating non-conventional renewable resources in Ecuador through GIS is detailed, highlighting that the areas with the most suitable solar radiation potential correspond to the locations in the Andes Mountains and the Galapagos Islands. However, the contribution generated has not been validated with surface measurements. Likewise, in [25] the updating of the GHI of the Solar Atlas of Ecuador is proposed. The information collected from the National Solar Radiation Database (NSRDB) of the United States, through the National Renewable Energy Laboratory (NREL), is validated and contrasted with surface measurements from three meteorological networks in the country. The results obtained show that 91% of a total of 54 stations analyzed can only use monthly information from satellites reliably, compared to 12% that can only use hourly profiles.

Solar irradiance data from the NREL database may be used for simulation purposes in Ecuador, due to the spatial and temporal coverage of its geostationary satellites. A physical solar model (PSM) [5] has been developed by NREL, which, through remote sensing, collects information regarding cloud properties, atmospheric profiles, aerosol properties, and surface and snow albedo, to produce a fast all-sky radiation model for solar application purposes (FARMS). However, the high cloud cover present on the Ecuadorian coast generates serious estimation problems in the model, due to absorption and multiple scattering within the clouds. Therefore, before using NREL data, it is essential to identify areas of high cloudiness index for further site-adaptation.

Table 1 summarizes information retrieved from 23 meteorological stations of the National Institute of Meteorology and Hydrology (INHAMI) [26], located at the highest altitude sites in the three Ecuadorian regions, plus the island region, where the Galápagos Islands are located. The sign criteria for latitude and longitude were negative south and west, respectively. Additionally, they were positive for north and east. The coastal region has an average cloud cover of about 84%, whereas the provinces with the lowest average cloud cover are Esmeraldas and Manabí. The Sierra region has an average cloud cover of 72%, with the provinces of Bolivar and Pichincha having the lowest cloud cover. The east region has an average of 73%, with Morona Santiago and Sucumbíos being the provinces with the lowest cloud cover. Finally, the Galapagos Islands have an average cloud cover of 75% for Santa Cruz Island. Therefore, the region with the highest percentage of cloud cover is the Ecuadorian coast, followed by the island, east and Sierra regions, respectively.

**Table 1.** Annual average of cloudiness index in various stations in Ecuador.

| Region | Station | Province | Latitude | Longitude | Altitude (m) | Cloudiness% |
|---|---|---|---|---|---|---|
| Coastal | Zaruma | El Oro | −3.6988 | −79.6113 | 1100 | 87.5 |
| | Quininde(Conv.Madres Lauritas) | Esmeraldas | 0.3194 | −79.4333 | 115 | 75.0 |
| | Guayaquil U.Estatal (Radio Sonda | Guayas | −2.2000 | −79.8833 | 6 | 87.5 |
| | Milagro (Ingenio Valdez) | Guayas | −2.1155 | −79.5991 | 13 | 87.5 |
| | Pichilingue | Los Ríos | −1.1000 | −79.4616 | 120 | 87.5 |
| | Julcuy | Manabi | −1.4800 | −80.6322 | 263 | 75.0 |
| | Santa Elena-Universidad | Santa Elena | −2.23333 | −80.9083 | 13 | 87.5 |
| | Puerto Ila | Santo Domingo | −0.4761 | −79.3388 | 319 | 87.5 |
| Sierra | Chanlud | Azuay | −2.6766 | −79.0313 | 3336 | 87.5 |
| | Laguacoto | Bolivar | −1.6144 | −78.9983 | 2622 | 62.5 |
| | Cañar | Cañar | −2.5519 | −78.9452 | 3083 | 75.0 |
| | El Angel | Carchi | 0.6263 | −77.9438 | 3000 | 75.0 |
| | Totorillas | Chimborazo | −2.0150 | −78.7222 | 3210 | 75.0 |
| | Cotopaxi-Clirsen | Cotopaxi | −0.6233 | −78.5813 | 3510 | 87.5 |
| | La Argelia-Loja | Loja | −4.0363 | −79.2011 | 2160 | 75.0 |
| | Illiniza-Bigroses | Pichincha | −0.6227 | −78.6594 | 3461 | 50.0 |
| | Calamaca Convenio Inamhi Hcpt | Tungurahua | −1.2761 | −78.8188 | 3402 | 62.5 |
| East | Macas San Isidro-Pns | Morona Santiag | −2.2102 | −78.1613 | 1110 | 50.0 |
| | Papallacta | Napo | −0.3650 | −78.1447 | 3150 | 87.5 |
| | San Jose De Payamino | Orellana | −0.5038 | −77.3175 | 345 | 75.0 |
| | Lumbaqui | Sucumbios | 0.0405 | −77.3338 | 580 | 62.5 |
| | Yanzatza | Zamora Chinchi. | −3.8375 | −78.7502 | 830 | 87.5 |
| Galapagos Islands | Bellavista-isla s.cruz | Galápagos | −0.7000 | −90.3666 | 194 | 75.0 |

The objective of this study is the validation and further site-adaptation of the NREL GHI for high cloud cover locations but whose methodology may be adaptable to other sites. The validation considered a parametric and non-parametric analysis of surface GHI measurements against NREL satellite data. For this purpose, surface and satellite GHI data were debugged. Two profiles were defined: hourly and monthly. Then, an exploratory and comparative analysis of the two data sets was performed. The parametric and non-parametric analysis identified the profile that could not be used due to its variability; and site-adaptation based on solar geometry, experimental measurements and NREL variables were developed. A cloud index hypothesis for different sky types was proposed. Linear and non-linear regressions and the use of statistical learning were applied.

## 2. Materials and Methods

This article considered the generation of a model capable of reducing the NREL GHI estimation errors in areas of high cloud cover, mainly on the coast regions close to uplands. Figures 1 and 2 show the general and detailed scheme of the five development phases considered. The first stage was based on verification and debugging of surface and satellite GHI data. The goal was to remove outliers in each data set by comparing surface and satellite GHI variables.

The second stage considered the application of parametric and non-parametric statistics for hourly and monthly GHI profiles. This phase aimed to validate the satellite data using surface measurements and the GHI profile which did not comply with the validation tests was selected for further adjustment. In the third stage, the input variables to the adjustment model were defined. The objective in this phase was to select predictor variables, based on the solar geometry and variables present in NREL, through Pearson's correlation analysis. The fourth stage involved the site-adaptation through the generation of the model with the information obtained in the second and third stages and then the GHI profile was divided into different sections according to the clearness index Kt. After this process, linear, non-linear regressions and the application of statistical learning through "Best Subset Selection" and "Forward Stepwise Selection" algorithms [27] were employed. Finally, the fifth phase evaluated the best-fit model for any given sky condition using statistical tests. The objective was to obtain a model capable of improving the correlation of the satellite data with respect to the initial surface measurements. Each of the five phases is described in detail in the following subsections.

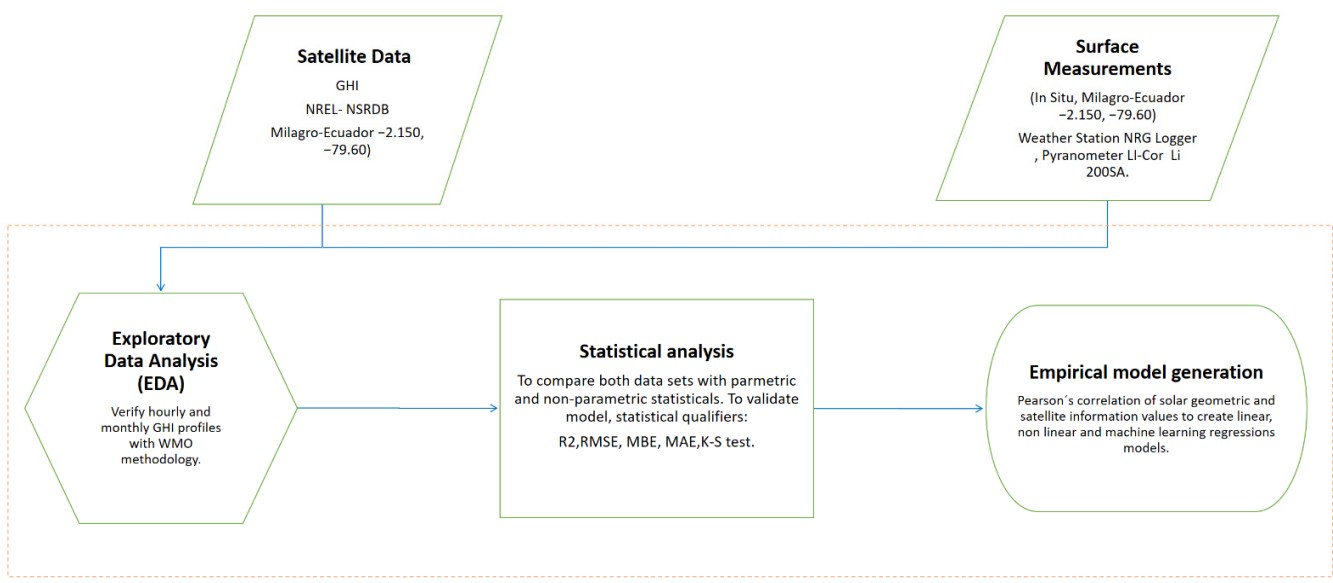

**Figure 1.** Methodology, the general scheme.

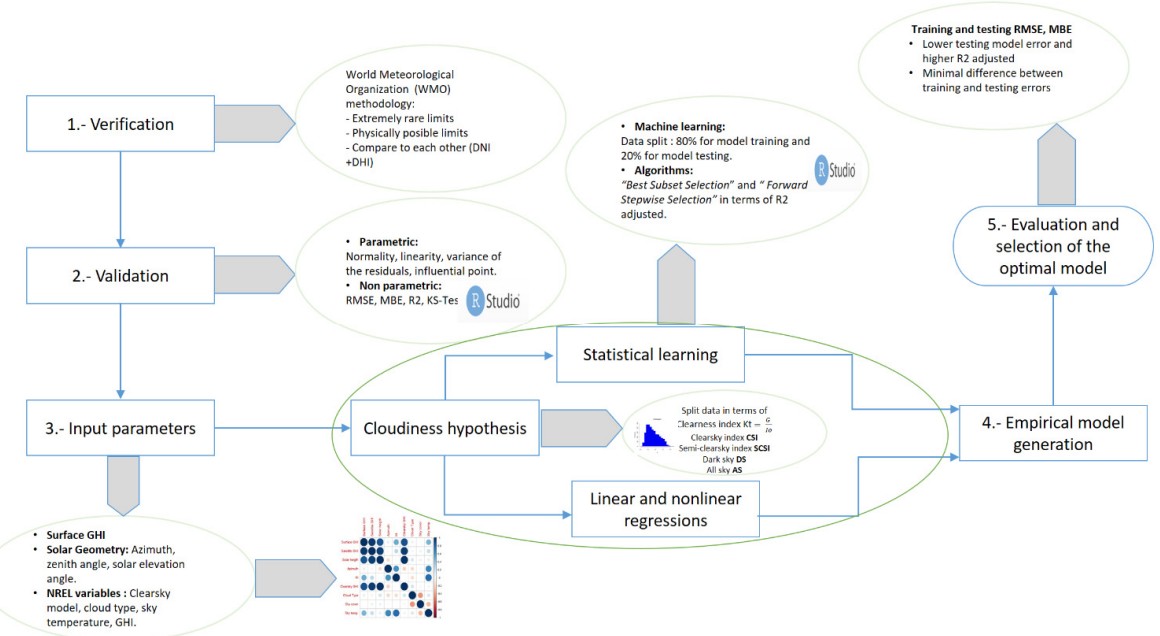

**Figure 2.** Methodology, specific scheme.

## 2.1. Surface Measurement Set Up

Experimental GHI data were retrieved from the meteorological station of the Universidad Estatal de Milagro (−2.150, −79.60) Ecuador, with a zenith angle at 12:00 of 72.77°, located on the coast at 13 m above sea level, with a humid tropical climate, where two seasonal periods can be distinguished: one rainy and the other dry. The rainy season is from December to May, while the dry season lasts from June to November. The annual average values are as follows [26]: temperature 26 °C, rainfall 912 mm, relative humidity 77%, wind speed 1.3 m s⁻¹ in a SW direction. The average annual cloudiness index corresponds to a value of 7 octas (87.5%).

The radiometric technology used was an LI-Cor Li 200S pyranometer with a measurement uncertainty of ±3% and data records every 15 minutes. The NRG logger was used for data acquisition and Symphonie Data Retriever (SDR) software for data processing [28]. Estimation of the satellite solar radiation, from the NREL database [3] was used for the

same location as surface measurements with spatial and temporal resolution of 2 km and 30 min, respectively. The satellite GOES 17 position was $-0.02$, $-75$. The relative position of the radiometric station to the satellite image pixel was $-8.52 \times 10^{-4}$, $-3.995 \times 10^{-3}$. The software used for data filtering, statistical analysis, and modeling was R Studio 4.1.1 2021 version. Surface GHI measurements from 2015 were used due to the higher quality of the data after an exploratory analysis from 2013 to 2017.

### 2.2. Data Processing

Once the surface and satellite GHI data were obtained, it was necessary to debug them. The surface data flags considered anomalous were identified using the SDR tool. The methodology proposed by the World Meteorological Organization (WMO) [29] was used for their treatment, where the following sequential criteria were highlighted:

- Physically possible limits;
- Extremely rare limits;
- Compared to each other (direct and diffuse irradiance).

The procedure and calculation expressions for data conditioning were described in [30], where the physically possible values corresponded to the following equation:

$$-4 \, \text{Wm}^{-2} < G < \varepsilon * Isc * 1.5 * (\cos\theta z)^{1.2} + 100 \, \text{Wm}^{-2} \tag{1}$$

where G represents the global horizontal irradiance, $\varepsilon$ is the terrestrial eccentricity, Isc is the extraterrestrial radiation, and $\theta z$ corresponds to the zenith angle.

Additionally, extremely rare limits, which exclude night hours from the analysis, were identified by the following expression:

$$-2 \, \text{Wm}^{-2} < G < \varepsilon * Isc * 1.2 * (\cos\theta z)^{1.2} + 50 \, \text{Wm}^{-2} \tag{2}$$

Since the radiometric station used as a base in this study does not have the instruments to measure direct and diffuse irradiance, the sum of these variables has not been compared with the global irradiance *G*.

Likewise, for the recovery of missing data, interpolation of the known measurements at their upper and lower limits within the same data series was considered [31].

Once the debugging process was completed, for the surface and satellite GHI data series, two profiles were disaggregated: an hourly and a monthly profile for each series. For the monthly profile, the data set was grouped according to time as a set of average daily values.

### 2.3. Parametric and Non-Parametric Satellite GHI Validation

Based on the hourly and monthly GHI profiles, a linear regression model was generated, whose dependent variable corresponded to the satellite GHI estimates, and the independent variable was represented by the surface GHI measurements. In this study, the parametric validation of linearity, analysis of medians, normality of residuals, the variance of the residuals, and identification of influential points [32] were verified in the model. The purpose of the application of these parameters was to identify the major differences between the two: surface and satellite GHI data were compared. Table 2 summarizes the parameters used, the tests performed, the scores, and the P values. Some of the parameters considered required only a graphical analysis to understand their behavior.

**Table 2.** Model assumptions considered.

| Parameter | Test, Graph | Score | *p*-Value |
|---|---|---|---|
| Linearity | Scatterplot, correlation R | >0.5 | - |
| Analysis of medians | Boxplot, graph | - | - |
| Normality | KS, histograms, Q-Q plot | - | >0.05 |
| The variance of the residuals | Breusch Pagan | - | >0.05 |
| Influential point | Leverage, Cook´s distance | ≥1 | - |

For the non-parametric validation of the hourly and monthly GHI series, statistical indicators of dispersion, such as the RMSE and MBE, were considered. The KS test, as a non-parametric indicator of model fit, was used to evaluate whether both data series come from the same distribution [33]. The expressions (3), (4) and (5) were used to calculate these indicators:

$$RMSE = \sqrt{\frac{\frac{1}{n}\sum_{i=1}^{n}\left(x_{mpi} - x_{sfi}\right)^2}{\frac{1}{n}\sum_{i=1}^{n}x_{sfi}^2}} \times 100 \tag{3}$$

where n represents the number of observations, $x_{mpi}$ refers to the predictions of the I-sth model element, and $x_{sfi}$ to the values of the I-sth term obtained with the surface measurements.

The indicator of the Root Mean Squared Error was set as a percentage for easier interpretation. Likewise, for the Mean Bias Error, the equation in (4) was used:

$$MBE = \frac{\frac{1}{n}\sum_{i=1}^{n}\left(x_{mpi} - x_{sfi}\right)}{\frac{1}{n}\sum_{i=1}^{n}x_{sfi}} \times 100 \tag{4}$$

On the other hand, the KS-test statistic is a test that considers how close the probability of a distribution drawn from a sample is to the probability of the reference distribution, which in this case are the surface measurements. The KS-test uses the distance D as an indicator, see Equation (5).

$$D = |(xsfi) - Fa(xmpi)| \tag{5}$$

where $F(xsfi)$ is the i-th term cumulative distribution function for surface GHI measurements and the function $F(xmpi)$ represents the cumulative distribution function of the I-sth term in the model predictions.

### 2.4. Site-Adaptation, Definition of Input Parameters

The parametric and non-parametric validation allowed the identification of the GHI profile that cannot be used for simulation purposes because of its high variability. From the identified GHI profile, the input variables to the model were defined. This work considered the solar geometry and NREL variables, which were chosen based on common graphical patterns of relationship with the surface GHI and with each other. The shape patterns that followed a linear or exponential trend were the clear sky model GHI, cloud type, sky cover and sky temperature. Table 3 details the definition of the predictors considered and the response variable, in this case, defined as the satellite GHI fit.

**Table 3.** Model predictors considered.

| Variable | Parameter | Source | Type |
|---|---|---|---|
| GHI | Surface GHI | Meteorological station | Input |
| $\alpha$ | Solar elevation angle | Solar geometry | Input |
| $\psi$ | Azimuth | Solar geometry | Input |
| $K_t$ | Kt | Solar geometry | Input |
| $CS_G$ | Clearsky GHI | NREL database | Input |
| Ct | Cloud type | NREL database | Input |
| Sc | Sky cover | NREL database | Input |
| St | Sky temperature | NREL database | Input |
| $GHI_{Sat}$ | Satellite GHI | NREL database | Output |

For the solar geometry variables, the model based on the clearness index Kt described in [34] was considered, which estimates the GHI from surface measurements, and is expressed as:
where,

$$0 \leq Kt \leq 1 \quad Kt = \frac{G}{Io} \tag{6}$$

where the variable I0 corresponds to the extraterrestrial radiation [35], which is given by:

$$Io = Ics\ \varepsilon\cos\theta z \tag{7}$$

where the term *Ics* refers to the solar constant; for this case, a value of 1367 W m$^{-2}$ was assumed [36]; $\varepsilon$ corresponds to the eccentricity of the Earth's orbit; and $\theta z$ represents the zenith angle, which was calculated as the angle complementary to the solar altitude ($\alpha$), from the horizontal plane of the observer.

The eccentricity of the Earth's orbit was calculated by:

$$\varepsilon = 1 + 0.033\ \cos(2\pi Jd/365.24) \tag{8}$$

the variable Jd corresponds to the day of the year, also called Julian day, where Jd $\in$ [1, 365]. Finally, the solar elevation angle ($\alpha$) was obtained from [37].

Correlation Analysis

With the selected input variables, a Pearson correlation diagram was performed. Figure 3a shows the level of association of the variables for the hourly profile and Figure 3b shows the level of association for the monthly profile. The level of association is represented by the color and the diameter of the circumference; the darker the blue, the higher the direct association, and the darker the red, the higher the inverse association. The greater the diameter of the circumference, the stronger the correlation. The information presented in both profiles made it possible to identify significant differences and to discard variables with low or no correlation. Thus, the greatest correlated predictor variables were identified to combine them in linear regressions that gave rise to heuristic models. The most appropriate variables were selected based on the criteria described in Section 2.5.1.

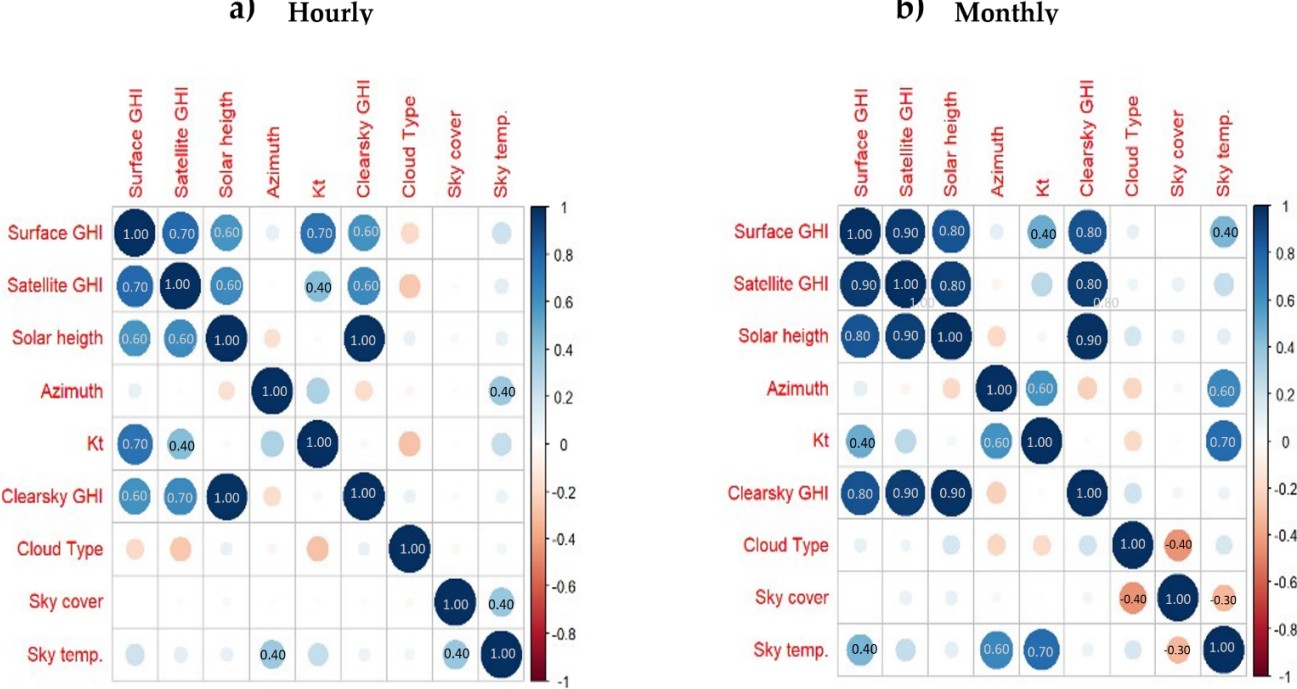

**Figure 3.** Pearson's correlation matrix, comparison between hourly and monthly GHI profiles. (**a**) Shows the hourly profile. (**b**) Shows the monthly profile. The greater diameter circles with darker shades show a better correlation. Blue, red and white colors represent direct, inverse and null correlation respectively.

### 2.5. Satellite GHI Fit Model Generation

With the input variables defined, a cloudiness hypothesis was made for different sky conditions, as a function of the clearness index Kt; unlike what is proposed in [38], the data series was split into four intervals: clear sky model (CSI), semi-clear sky (SCSI), cloudy sky (DS), and all-sky conditions (AS). These intervals were derived from the analysis of the density distribution and the empirical criteria of the authors of this paper, where values of Kt > 0.7 were proposed for the first CSI section. For the second section, values between 0.5 < Kt < 0.7 were set for SCSI, and values of Kt < 0.5 were used in the DS model. Finally, for AS, Kt was used within its full range of information. Figure 4 shows the distribution of the Kt index for the hourly GHI profile. Furthermore, each Kt stretch was separated into two data sets: training data and test data. The training data were randomly generated and corresponded to 80% of the total stretch, while the test data corresponded to the remaining 20%.

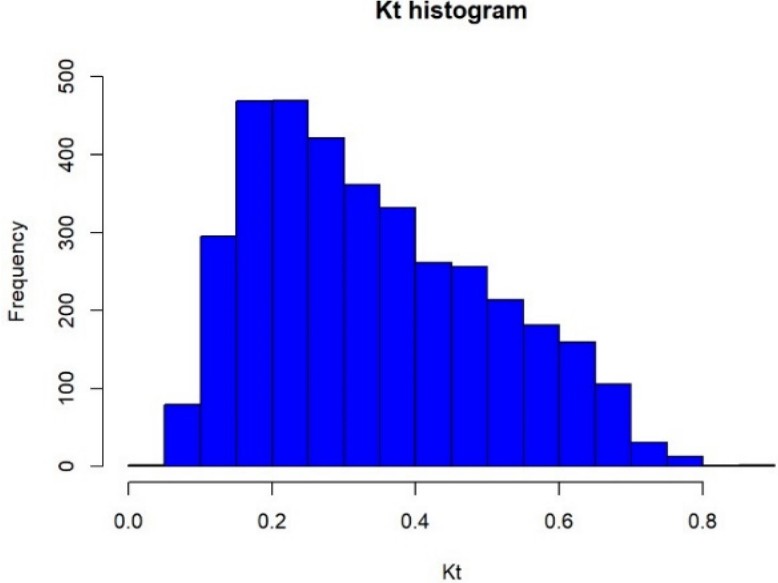

**Figure 4.** Kt index distribution.

### 2.5.1. Application of Linear, Non-Linear Regressions and Machine Learning

Once the best correlating variables and the Kt sections were segmented into training and test data, different combinations of predictors were tested. The computer tool used allowed the generation of several heuristic models to obtain the correction of the GHI satellite model. Linear, non-linear and polynomial regressions were applied. The selection criteria for the heuristic regression models were the F-statistic and the adjusted $R^2$. The F-statistic is a test that indicates the overall significance of the model. Models with a *p*-value < 0.05 were considered valid. The adjusted $R^2$ indicates the degree to which the predictor or independent variables explain the dependent variable, in this case, the satellite GHI. Adjusted $R^{-2}$ > 0.5 was set for the model to be valid.

The application of statistical learning [39] using the machine learning algorithms "Best Subset Selection" and "Forward Stepwise Selection" [27] was selected to generate linear regression models. Unlike common linear models that use least squares regression, both algorithms have been used due to the following criteria: (a) number of predictors found, (b) cross-validation, which finds the optimal combination of predictors and avoids model overfitting, and (c) computing efficiency. The "Best Subset Selection" algorithm was selected to search for a model that uses an optimal number of predictor variables, without generating overfitting, and setting the model, with the highest adjusted $R^2$ as the iteration parameter. "Best Subset Selection" evaluates all possible models that can be generated by combining n predictor variables to select the best one. The sequential basis is as follows:

1. A null model $M_0$ with no predictors is generated.
2. Then, $M_1$ models containing only one predictor are generated and the best one is selected.
3. The previous process is repeated for models containing two variables and so on until the n predictor variables are completed.
4. From among all the models ($M_0$, $M_1$, $M_2$, ... $M_n$), the best one is selected through cross validation, which for this study will be the adjusted $R^2$.

Likewise, the "Forward Stepwise Selection" algorithm used the iteration of the combinations of each of the predictor variables to choose the most efficient one, based on cross-validation, it also considered the highest adjusted $R^2$. The main difference with the "Best Subset Selection" algorithm is its higher computational efficiency. The process of searching for the best model from a null model $M_0$ with no predictors is the same. Then, a predictor is introduced and the best $M_1$ is selected from the training errors. The same process is repeated by increasing the number of predictors until the model $M_n$ is reached. The best model is chosen this time from the last model generated, built from $M_{n-1}$. Although the Forward Step Selection algorithm does not evaluate all possible combinations; its computational cost is lower.

### 2.6. Selection and Evaluation of the Optimal Model

The regression models generated with the training data (80%) was contrasted against the test data (20%); therefore, the evaluation involved the comparison between the training RMSE and MBE errors versus the test RMSE and MBE errors. The expressions for calculating these errors are defined above in Section 2.3 (3) and (4). Models with a high adjusted $R^2$, low training RMSE, and lower predictors were considered suitable. The selection of the final optimal model, in addition to achieving the parameters defined above, was valid for any sky condition, and considered the entire Kt amplitude. The application of the correction model was performed verifying the increase in $R^2$ with respect to the initial satellite GHI estimates of the profile that did not comply with the validation process.

### 3. Results

#### 3.1. Verification

One year of GHI data was retrieved from the weather station. The data were aggregated into hourly profiles corresponding to the year 2015. Data for nighttime hours were deleted. Filtering applied detected outliers that were recovered by linear interpolation and which correspond to 6% of the total data. Thus, 30 values were not able to be interpolated due to the presence of 3 gaps in 10 different days. With this procedure, 4350 high-quality data were obtained. Analogous to the previous process, NREL outliers were identified and recovered by linear interpolation. No gaps were found in this data set. The physically possible NREL data were corrected, observing the relationship between the GHI and its components, as a result of the addition of the direct radiation plus the diffuse radiation. Then, 657 values were identified as not complying with the quality labels. Using the necessary interpolations within the same data set, 4350 quality data were obtained. Finally, hourly and monthly profiles were prepared before validation.

#### 3.2. Validation

The parametric analysis of linearity for surface GHI versus satellite GHI is shown in the scatterplots. Figure 5 shows the hourly profile and Figure 6 the monthly profile. For the monthly profile, the data set was aggregated as monthly average daily data, which did not consider nighttime hours. It can be seen that in both Figures 5 and 6, there is a linear correlation. In the case of Figure 5, the highest data density was in the order of 125 W m$^{-2}$, with a coefficient of determination $R^2$ of 0.607. In the case of Figure 6, the highest data concentration was around 350 W m$^{-2}$, where the coefficient of determination $R^2$ increased to 0.905, indicating that the monthly aggregations presented a better fit to the linear model compared to the hourly profile. The correlation output parameters of the

hourly profile were y = 1.02x + 93.63 and the output parameters of the monthly profile were y = 1.2x + 35.2.

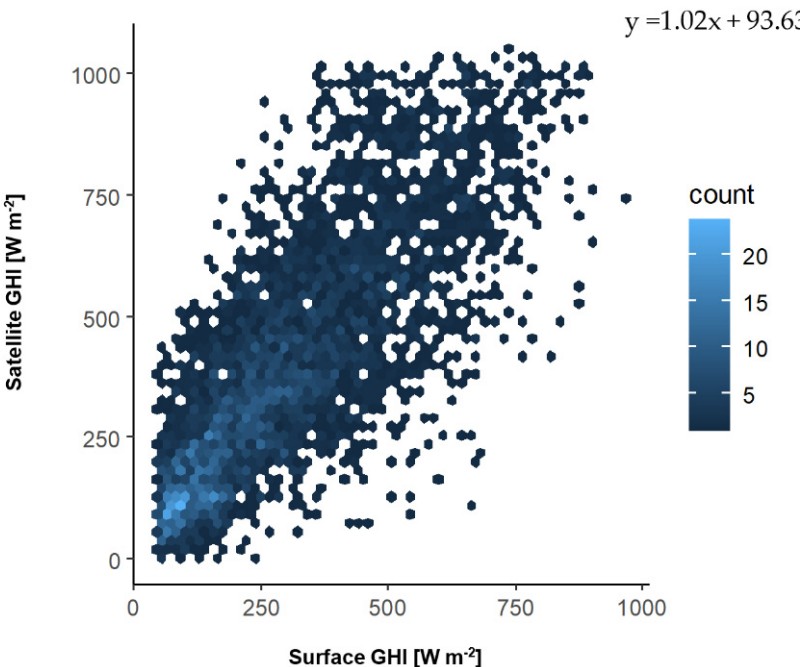

**Figure 5.** Scatterplot of hourly surface vs. satellite GHI profiles.

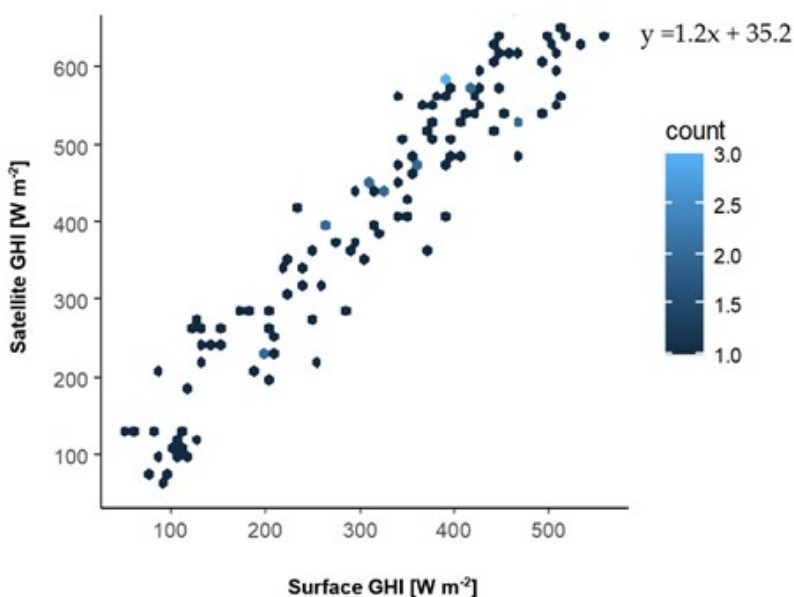

**Figure 6.** Scatterplot of monthly surface vs. satellite GHI profiles.

It was also possible to assess linearity through a point plot between the predictions of the GHI model versus the residuals. Figure 7 shows that while the volume of data in Figure 7a is larger, it is more difficult to distinguish any pattern or orderly behavior of the distributions in the hourly profile. Figure 7b shows a random distribution above and below the neutral value and so the existence of linearity in the hourly profile is verified.

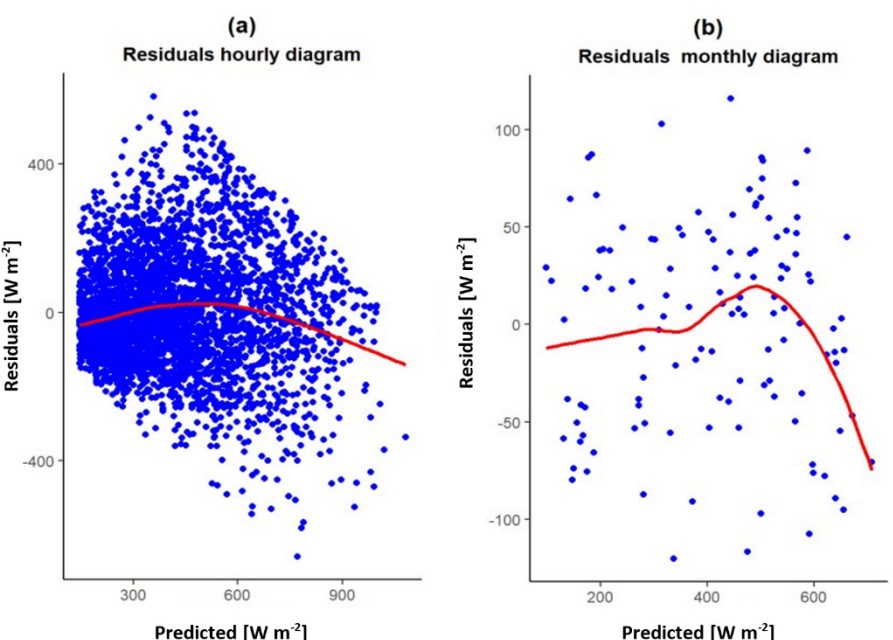

**Figure 7.** Scatterplots of normality test for hourly vs. monthly GHI profiles. Plot (**a**) represents the distribution of the predicted values versus residuals for the hourly profile. (**b**) shows the predicted values versus residuals for the monthly profile.

The parametric analysis of the medians in the boxplot of Figure 8 shows the differences in the hourly GHI profiles between surface and satellite data. The minimum values represented in the lower whiskers presented slight differences, due to the filtering of the series, in the order of 50 W m$^{-2}$. The first quartile of the boxplot from the bottom whisker to the box reached values between 180 W m$^{-2}$ and 200 W m$^{-2}$ for the surface and satellite GHI, respectively. The median value was 300 W m$^{-2}$ for the surface and 400 W m$^{-2}$ for the satellite. The value for the third quartile corresponded to a value of 450 W m$^{-2}$ for the surface and 600 W m$^{-2}$ for the satellite. Finally, the upper whisker reached 825 W m$^{-2}$ for the surface with normal presence of outliers and 1030 W m$^{-2}$ for the satellite data. It can be noted that the satellite GHI overestimates the surface GHI for the Ecuadorian coastal regions, in agreement with the results obtained in [24].

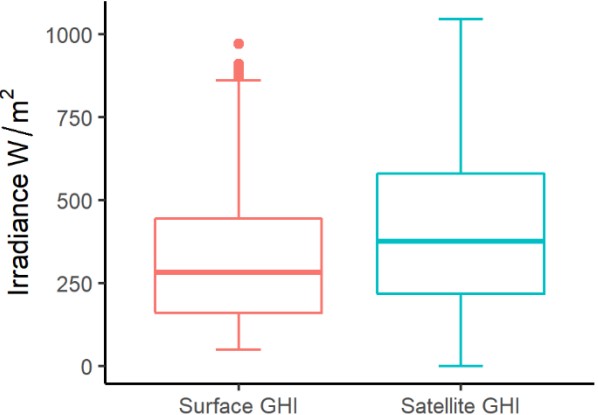

**Figure 8.** Boxplot comparison between surface and satellite GHI estimation [W m$^{-2}$].

Figure 9 presents the annual analysis between surface GHI measurements compared to satellite GHI estimation. In the case of surface GHI, there was a greater variation throughout the year, with medians between 250 W m$^{-2}$ and 375 W m$^{-2}$. In the case of satellite radiation, the medians vary in the range of 300 W m$^{-2}$ to 400 W m$^{-2}$. The outlier values did not

exceed 950 W m$^{-2}$ and 1100 W m$^{-2}$ for surface and satellite GHI, respectively. Throughout the year, the irradiance measured from the surface as well as the irradiance estimated by the satellite presented a homogeneous behavior because the data were obtained from the equatorial zone. It can be seen that month by month, the behavior of the satellite GHI continued to be overestimated.

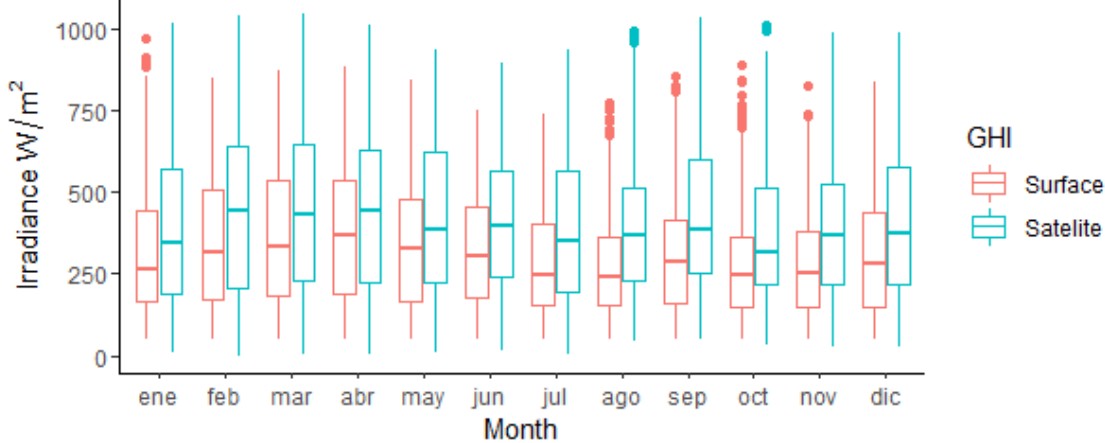

**Figure 9.** Comparison boxplot between surface and satellite GHI estimations, hourly profile.

Figure 10 presents the monthly GHI profile between surface measurements and satellite data. For the surface GHI, the variation in the medians was in the order of 340 W m$^{-2}$ to 400 W m$^{-2}$. For the satellite GHI, the variation in the medians was in the order of 420 W m$^{-2}$ to 500 W m$^{-2}$. In both GHI profiles, the absence of higher outliers can be observed due to the aggregation of the data.

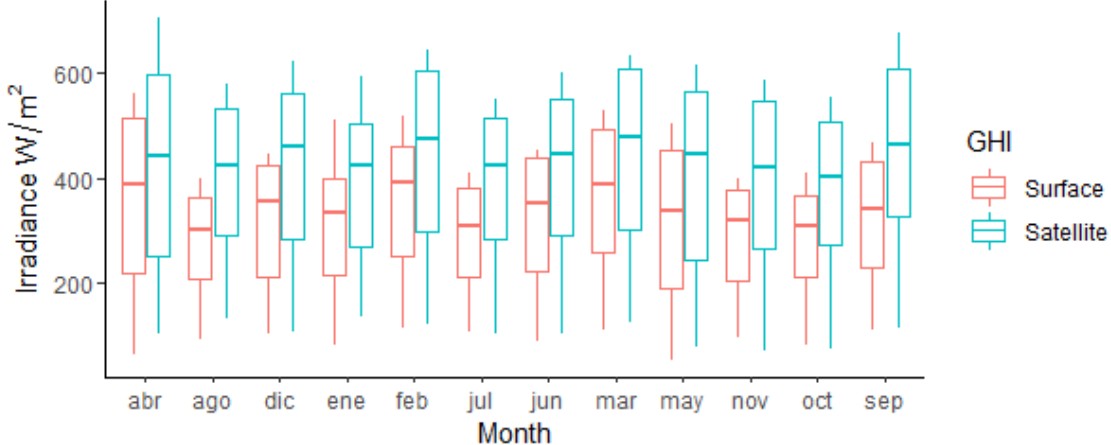

**Figure 10.** Comparison boxplot between surface and satellite GHI estimations, monthly profile.

For the normality analysis, Figure 11 presents the comparison of the hourly profile distributions for the surface GHI data with the GHI data obtained from satellite images. In both cases, the distributions presented asymmetry in their fitting curves, and did not correspond to a normal distribution since the mean and median are not at the neutral point. However, with a deeper analysis, it is possible to identify the temporal sectors, and then perform a segmentation that contributes to the elimination of clusters, which causes the shape of the distribution to deviate from the normal shape. This process is called Quantile Mapping (QM). Although, this technique is out of the scope of this study.

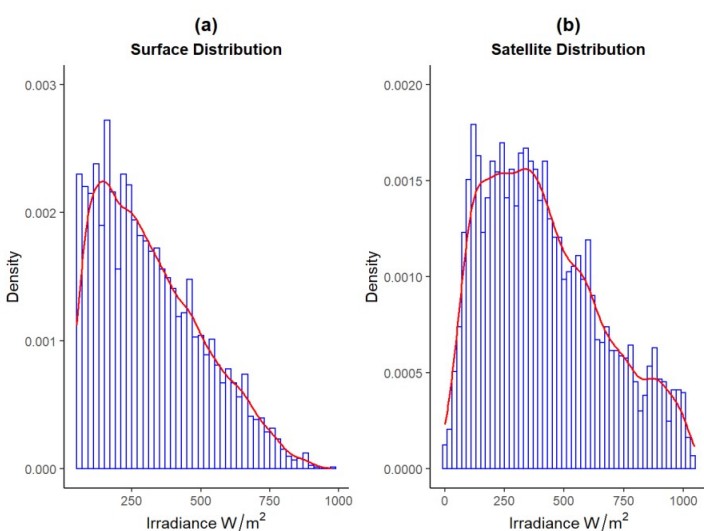

**Figure 11.** Histogram of frequency for hourly GHI profile. (**a**) Shows the irradiance distribution for the surface measurements. (**b**) Shows how the irradiance of the satellite estimates is distributed.

Figure 12 shows the comparison of the distributions of the model residuals for hourly and monthly GHI profiles. Figure 12a represents a leptokurtic kurtosis, while the monthly distribution in Figure 12b evidences a right skewed distribution. Therefore, a normal distribution in the residuals can be interpreted as suggesting the presence of a lower number of outliers, which is in agreement with the debugging process of both data series performed previously.

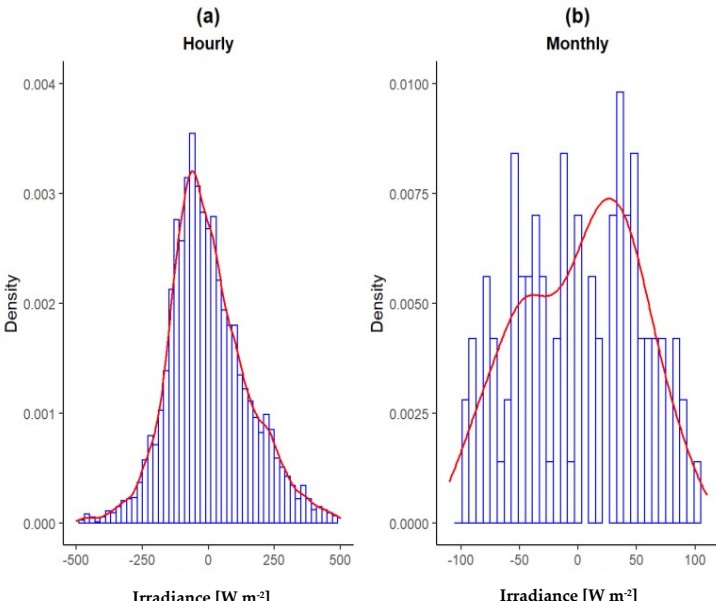

**Figure 12.** Histogram of residuals frequency for hourly vs. monthly GHI profiles. (**a**) Shows the irradiance distribution for the hourly profile. (**b**) Shows how the irradiance is distributed for the monthly profile.

In [40], different types of normality tests, which are performed as a function of the number of observations, and which evaluate the nature of the information are established. The KS-test was chosen to determine the normality of the surface and satellite GHI distributions because both the hourly and monthly profiles represent more than 50 observations in each dataset. The tests performed considered a null hypothesis of normality H0 which is accepted if the *p*-value is greater than 0.05. For the hourly surface and satellite profiles,

the *p*-value was $<2.2 \times 10^{-16}$ for both cases. The KS-test for the monthly profiles showed a *p*-value of 0.02 for the surface data and a *p*-value of 0.003 for the satellite data.

Figure 13 represents the quantile–quantile plot of the comparison between the hourly analysis Figure 13a and the monthly analysis Figure 13b. The x-axis represents the theoretical survival probability of the data and the y-axis the empirical or sample probability of the GHI data. The monthly profile of the blue points follows a normal distribution represented by the red solid line, due to the smaller amount of aggregated data. Using this graphical method, it is possible to determine whether the GHI data distribution is normal, in the case where the blue points coincide with the red line. However, the values were misaligned from the reference line, mainly at the extremes of both graphs, suggesting the presence of tails in the distribution.

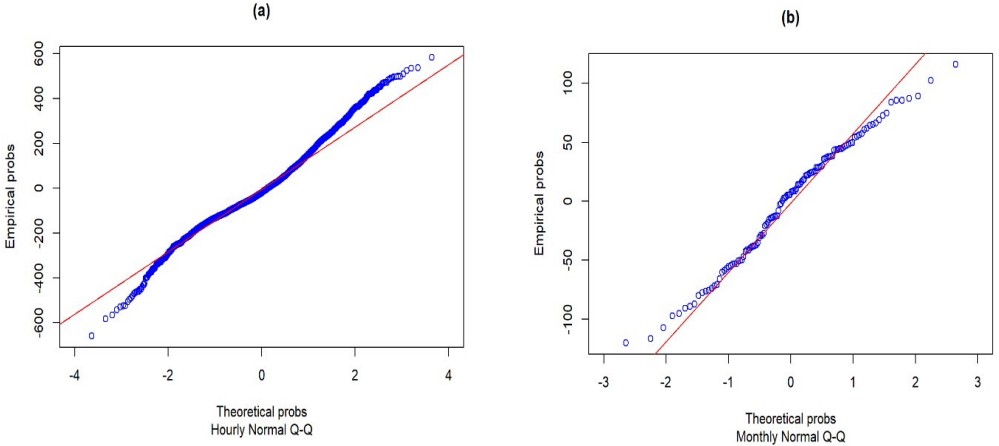

**Figure 13.** Normal Q-Q Plot, comparison between hourly vs. monthly GHI profiles in [W m$^{-2}$]. (**a**) Shows the probability of survival of the GHI data for the hourly profile, while (**b**) represents the probability of survival of GHI for the monthly profile.

Another assumption corresponds to the homogeneity of variance of the residuals, which was evaluated by applying the Breusch Pagan test. The null hypothesis, H$_o$, recognizes the existence of homoscedasticity, being true if the *p*-value is less than 5%. In the case of the GHI hourly profile, the *p*-value was $<2.22 \times 10^{-16}$. On the other hand, in the test carried out on the monthly GHI profile, a *p*-value of 0.87 was obtained. Figure 14a shows the hourly GHI model profile and Figure 14b the monthly GHI profile. The red line of the latter shows a more linear trend, where homoscedasticity is proved, corroborating the result of the Breusch Pagan test.

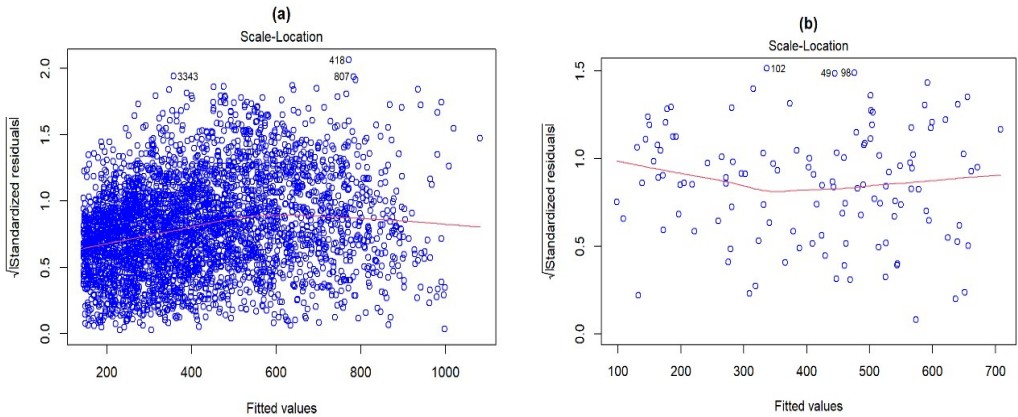

**Figure 14.** Homoscedasticity scatterplot of the hourly vs. monthly GHI profiles in [W m$^{-2}$]. (**a**) Shows the variance of the residuals for the hourly GHI profile, while (**b**) shows the variance of the residuals for the monthly GHI profile.

In the influential point analysis, the surface or satellite GHI data with large differences in values (outliers) have a significant influence in the generation of a model. The leverage effect affects the coefficients of the linear equation leading to prediction errors. For this reason, it is necessary to identify the influential values of hourly and monthly GHI profiles. In this study, Cook's distance is considered an indicator of influence, being greater than 1 for those values that represent an effect on the model. Figures 15 and 16 represent the observations with Cook's distance, both for the hourly and monthly GHI profiles, respectively. The number labels in Figure 15a,b indicate the position of the most influential GHI data; however, it can be seen that on the scales, none of these values exceed 1, thus demonstrating the correct data-filtering methodology was applied. In Figure 16a,b, none of the points exceeded Cook's geometric limit, which is just visible in the lower left corner.

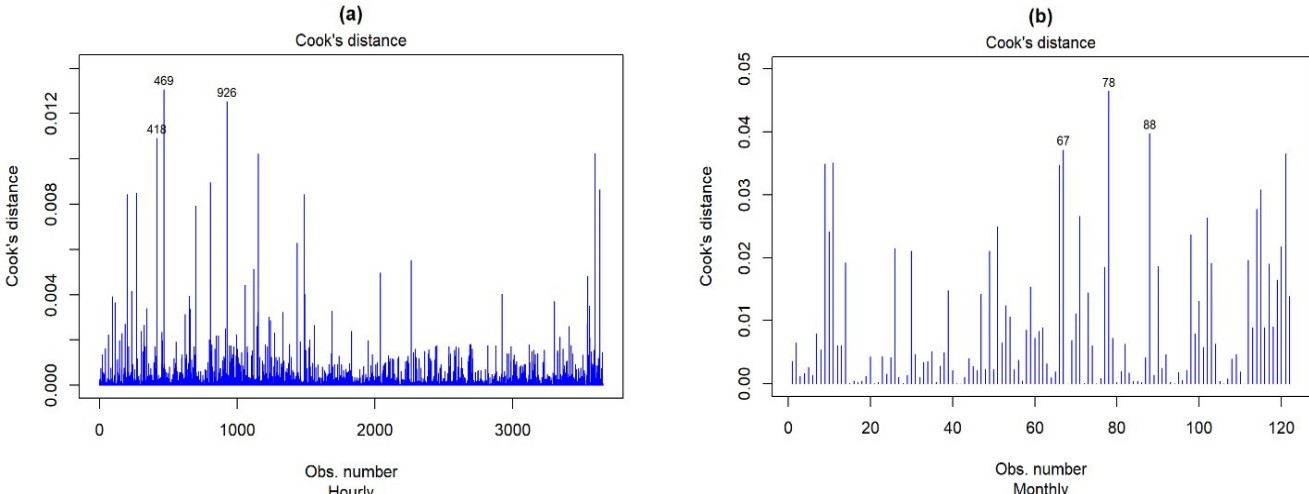

**Figure 15.** Cook's distance plot between hourly vs. monthly profiles. (**a**) Shows the positions of the influence values for the hourly GHI profile. (**b**) Shows the positions of the influence points for the monthly GHI profile.

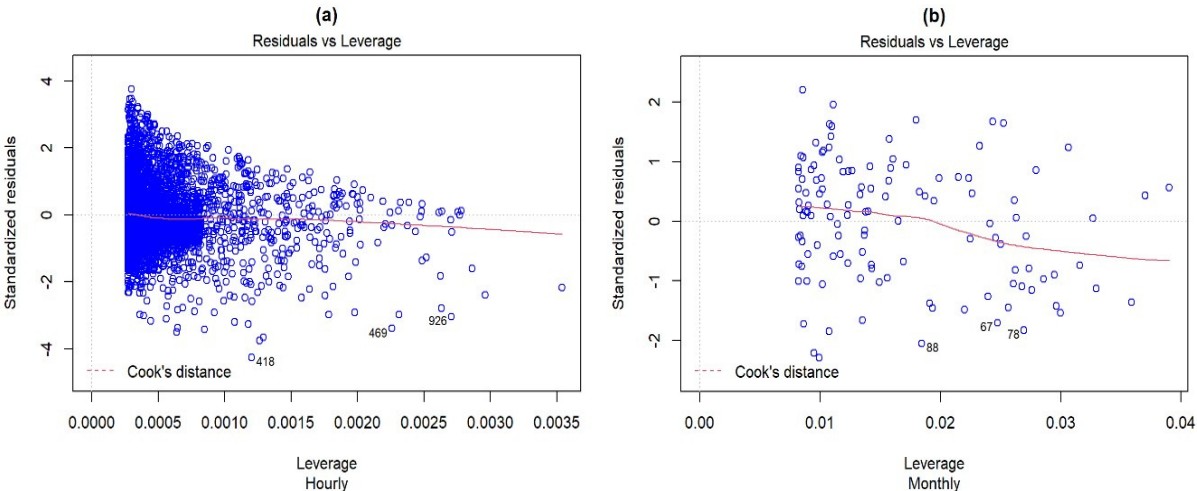

**Figure 16.** Cook's distance of residuals versus leverage between hourly vs. monthly GHI profiles. (**a**) Shows the Cook's distance for the hourly profile. (**b**) Shows the Cook's distance for the monthly profile.

The parametric analysis reveals that some of the tests cannot be used to validate the hourly profile, because of the nature of the data and the greater variation in the hourly satellite GHI profiles, attributed to the high cloudiness of the Ecuadorian coast. The non-parametric analysis summarized in Table 4 represents the results obtained from the GHI data series for the hourly and monthly profiles. While the coefficient of determination $R^2$ for the monthly profile increases with respect to the hourly profile, the model errors RMSE and MBE also increase slightly, which, being positive, indicate an overestimation for the satellite estimations. However, considering for the KS-test a null hypothesis Ho, which contemplated normality between both data series, with a significance of 0.05, only the monthly profile complied with this parameter.

**Table 4.** Non-parametric validation for hourly and monthly irradiance profiles.

| Model | $R^2$ | RMSE % | MBE % | KS-test *p*-Value |
|---|---|---|---|---|
| Hourly | 0.607 | 26.79 | 31.17 | <0.0000001 |
| Monthly | 0.905 | 29.93 | 31.40 | 0.142 |

*3.3. Site-Adaptation and Cloudiness Model*

The results of the parametric and non-parametric validation identified the hourly GHI profile as not valid for its application in simulation, because of its high temporal variation. The site-adaptation was considered the split of the hourly profile into different sections, corresponding to clear, semi-clear, and cloudy skies. It was found that for the CSI model, there are only 1.29% of observations in cloud-free conditions. For the SCSI scenario, 18% and, for the DS model, 80.7%. This corroborates that the average cloudiness on the Ecuadorian coast is above 80%, which may generate important estimation errors when using satellite images.

The site-adaptation has given rise to the generation of different models, which range from the empirical to the heuristic, considering the correlation parameters, and which depend on the Kt section evaluated. Several combinations of predictor variables were generated. The models with the index 1 correspond to all types of sky, and integrate the eight variables selected. Models from index 2 to 5 belong to heuristic combinations based on the best correlation. Index 6 uses second- and third-order non-linear models. Finally, index 7 and 8 models use statistical learning. As a result, 31 models were obtained for all sky conditions. Table 5 summarizes the parameters considered in the different models obtained. The F-statistic denoted the validity of each one. Low RMSE values indicated a good model fit. For the MBE, positive values denoted an overestimation of the model while negative values indicated an underestimation of the model. For both RMSE and MBE, random test samples were considered. A total of 31 models were generated. CSI models had the highest adjusted $R^2$. The best performers were the CSI 1, CSI 7 and CSI 8. CSI 1 used all predictors, and CSI 7 and CSI 8 employed statistical learning. The lowest performing models were CSI 4 and CSI 5. The best SCSI models were the SCSI 7, SCSI 8, SCSI 1 and SCSI 3, while the lowest performers were the SCSI 4 and SCSI 5. Note that the SCSI 7 and SCSI 8 models in this case share the same equation and the same statistical indicators. The DS 1, DS 7 and DS 8 models were the best performers, while the DS 4 and DS 5 were the low performers. The best AS models were AS 1, AS 7 and AS 3 and the worst performers were AS 4 and AS 5.

It is noticeable that non-linear models are not necessarily the most accurate. Additionally, it can be noted that if a model has a higher adjusted $R^2$, this does not mean that its model errors and biases are lower. A greater number of predictors in a model denotes lower errors because each will try to contribute to the response variable to predict its performance.

**Table 5.** Summary of generated models for all sky conditions.

| Model | Equation | $R^2$ Adjust | F Statistic | P Value | RMSE Testing % | MBE Testing % |
|---|---|---|---|---|---|---|
| CSI 1 | $220.535 - 1.415GHI + 41.068\alpha + 27.06\psi - 88.395K_t + 2.048CS_G - 12.768Ct - 0.326Sc + 0.289St$ | 0.889 | 38.88 | $9.21 \times 10^{-14}$ | 32.45 | 9.16 |
| CSI 2 | $235.119 - 1.251GHI - 114.444\alpha - 158.174K_t + 2.105 CS_G$ | 0.884 | 73.21 | $3.18 \times 10^{-16}$ | 32.25 | 10.74 |
| CSI 3 | $148.71 - 1.022GHI + 1.727CS_G - 13.641Ct$ | 0.899 | 113.2 | $<2.2 \times 10^{-16}$ | 30.1 | 10.13 |
| CSI 4 | $480.79 + 734.07\alpha - 678.06K_t$ | 0.881 | 141.2 | $<2.2 \times 10^{-16}$ | 38.65 | 14.3 |
| CSI 5 | $516.66 + 741.42\alpha + 19.24\psi - 767.38K_t$ | 0.88 | 94.06 | $<2.2 \times 10^{-16}$ | 38.55 | 13.32 |
| CSI 6 | $380.28 + 1661.56\alpha - 159.48\alpha^2 - 223.77\alpha^3$ | 0.896 | 110.7 | $<2.2 \times 10^{-16}$ | 32.12 | 10.06 |
| CSI 7 | $139.243 - 1.469GHI + 28.40\psi + 2.143CS_G - 12.825Ct$ | 0.901 | 87.41 | $<2.2 \times 10^{-16}$ | 31.13 | 9.69 |
| CSI 8 | $68.351 + 0.809CS_G - 16.205Ct$ | 0.894 | 161.2 | $<2.2 \times 10^{-16}$ | 26.8 | 8.25 |
| SCSI 1 | $57.413 + 0.419GHI + 275.76\alpha - 1.298\psi - 20.17K_t + 0.218CS_G - 27.118Ct - 0.107Sc - 1.575St$ | 0.824 | 309.9 | $<2.2 \times 10^{-16}$ | 20.3 | −0.448 |
| SCSI 2 | $-104.617 + 0.397GHI + 229.291\alpha + 127.25K_t + 0.294CS_G$ | 0.775 | 455.5 | $<2.2 \times 10^{-16}$ | 23.87 | −3.04 |
| SCSI 3 | $27.481 + 0.346GHI + 0.611CS_G - 27.249Ct$ | 0.823 | 817 | $<2.2 \times 10^{-16}$ | 19.56 | −0.454 |
| SCSI 4 | $-295.74 + 682.32\alpha + 516.36K_t$ | 0.766 | 863.2 | $<2.2 \times 10^{-16}$ | 25.8 | −2.861 |
| SCSI 5 | $-297.125 + 678.274\alpha - 2.981\psi + 528.261K_t$ | 0.765 | 574.9 | $<2.2 \times 10^{-16}$ | 25.78 | −2.868 |
| SCSI 6 | $550.397 + 5584.196\alpha - 655.037\alpha^2$ | 0.767 | 867.6 | $<2.2 \times 10^{-16}$ | 23.96 | −2.508 |
| SCSI 7 | $23.474 + 0.386GHI + 275.9010\alpha + 0.243CS_G - 27.452Ct$ | 0.825 | 623.4 | $<2.2 \times 10^{-16}$ | 20.33 | −0.448 |
| SCSI 8 | $23.474 + 0.386GHI + 275.9010\alpha + 0.243CS_G - 27.452Ct$ | 0.825 | 623.4 | $<2.2 \times 10^{-16}$ | 20.33 | −0.448 |
| DS 1 | $157.89 + 1.302GHI - 85.187\alpha - 0.839\psi - 369.4K_t + 0.175CS_G - 19.147Ct + 0.098Sc - 0.7955St$ | 0.631 | 504.1 | $<2.2 \times 10^{-16}$ | 37.72 | −0.131 |
| DS 2 | $96.677 + 1.34.GHI - 75.772\alpha - 332.604K_t + 0.128CS_G$ | 0.604 | 900 | $<2.2 \times 10^{-16}$ | 39.65 | 0.057 |
| DS 3 | $65.088 + 0.939GHI + 0.187CS_G - 19.02Ct$ | 0.627 | 1322 | $<2.2 \times 10^{-16}$ | 37.7 | −0.178 |
| DS 4 | $-173.757 + 362.933\alpha + 911.806K_t$ | 0.548 | 1430 | $<2.2 \times 10^{-16}$ | 41.12 | −0.373 |
| DS 5 | $-178.696 + 358.44\alpha - 8.763\psi + 944.08K_t$ | 0.552 | 967.9 | $<2.2 \times 10^{-16}$ | 40.94 | −0.12 |
| DS 6 | $386.561 + 8448.735GHI - 476.519GHI^2$ | 0.589 | 1692 | $<2.2 \times 10^{-16}$ | 40.82 | 0.275 |
| DS 7 | $155.66 + 1.31GHI - 87.78\alpha + 2.143K_t - 12.825Ct$ | 0.631 | 806.7 | $<2.2 \times 10^{-16}$ | 37.76 | −0.149 |
| DS 8 | $199.445 + 1.471GHI - 538.095K_t - 19.135Ct$ | 0.63 | 1339 | $<2.2 \times 10^{-16}$ | 37.76 | −0.194 |
| AS 1 | $142.8 + 0.991GHI + 43.85\alpha - 2.76\psi - 268.629K_t + 0.121CS_G - 23.007Ct + 0.013Sc - 0.276St$ | 0.697 | 840.3 | $<2.2 \times 10^{-16}$ | 34.45 | −0.527 |
| AS 2 | $81.165 + 1.087GHI + 31.408\alpha - 277.566K_t + 0.087CS_G$ | 0.662 | 1435 | $<2.2 \times 10^{-16}$ | 35.54 | −0.105 |
| AS 3 | $52.665 + 0.715GHI + 0.3CS_G - 23.127Ct$ | 0.692 | 2195 | $<2.2 \times 10^{-16}$ | 34.77 | −0.515 |
| AS 4 | $-185.903 + 452.876\alpha + 664.281K_t$ | 0.586 | 2073 | $<2.2 \times 10^{-16}$ | 39.23 | 0.088 |
| AS 5 | $-189.315 + 446.845\alpha - 9.407\psi - 694.068K_t$ | 0.59 | 1401 | $<2.2 \times 10^{-16}$ | 39.15 | 0.243 |
| AS 6 | $416.734 + 10468.523GHI - 1332.329GHI^2$ | 0.621 | 2398 | $<2.2 \times 10^{-16}$ | 37.56 | −0.283 |
| AS 7 | $137.629 + 0.986GHI - 2.755\psi - 264.258K_t + 0.18CS_G - 23.062Ct$ | 0.697 | 1345 | $<2.2 \times 10^{-16}$ | 34.46 | −0.511 |
| AS 8 | $168.98 + 1.236GHI - 487.002K_t$ | 0.659 | 2825 | $<2.2 \times 10^{-16}$ | 35.56 | −0.511 |

Figure 17 shows the results of the application of the "Best Subset Selection" algorithm, showing the number of variables as a function of the adjusted $R^2$ for each of the Kt sections. Note that it is not necessary to use the highest number of predictors to obtain the highest coefficient of determination which allowed some flexibility in the model in terms of the number of independent variables to used. For all models, the increase in adjusted $R^2$ was significant from the use of two predictors onwards. The CSI models achieved the highest adjusted $R^2$, followed by the SCSI models, the AS models and finally the DS models. A higher cloudiness index of the model denoted a lower adjusted $R^2$.

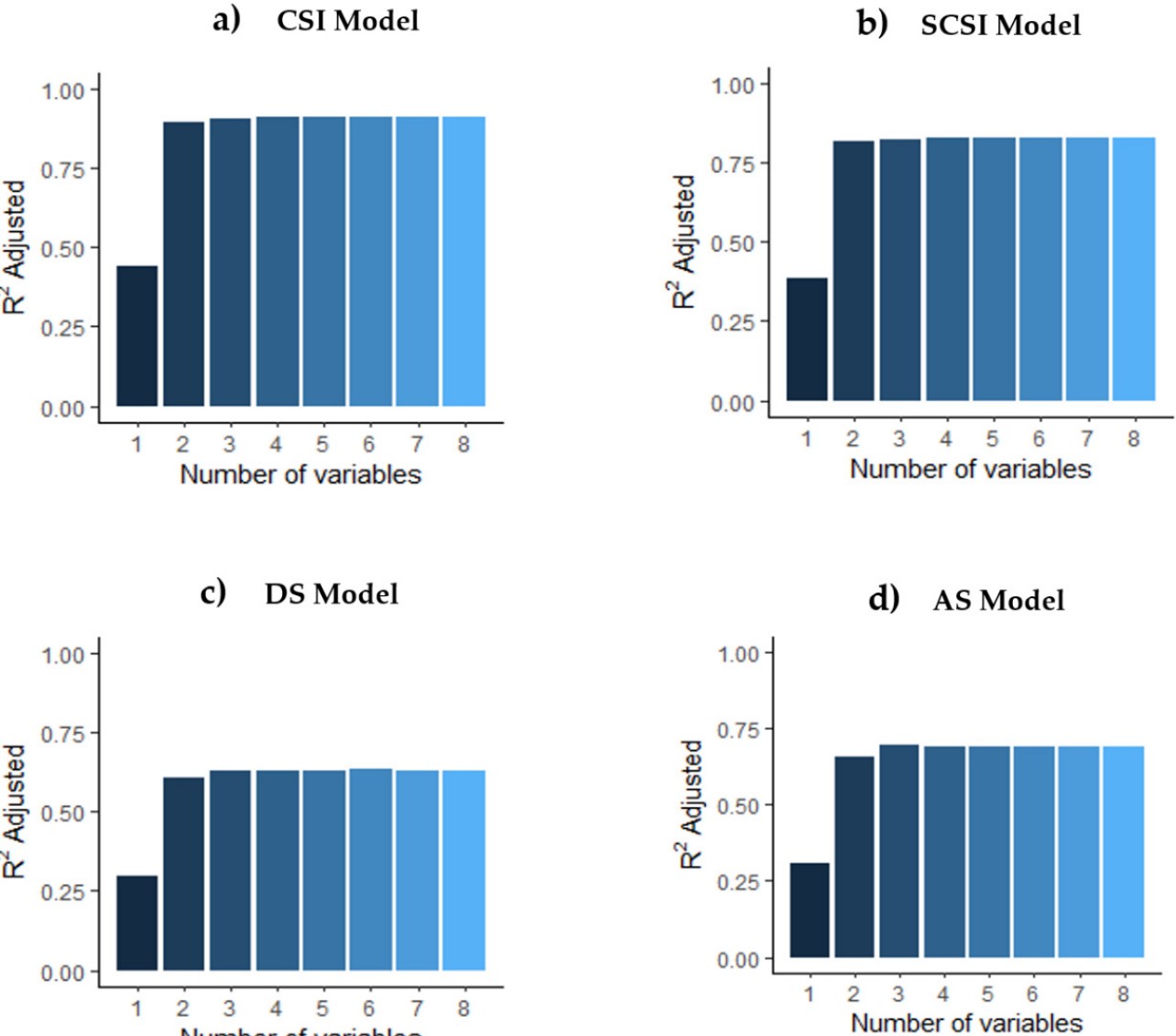

**Figure 17.** Best subset selection algorithm of number of variables for all Kt sections. The number of variables versus the adjusted R2 is shown in: (**a**) for the clear sky model CSI, (**b**) for the semi-clear sky model SCSI, (**c**) cloudy sky model DSI and (**d**) all sky conditions model AS.

Figures 18 and 19 present the results of the model RMSE errors and MBE bias errors of test samples for each sky type, including AS, which uses the entire Kt range. Figure 18, represented by the testing RMSE model errors, increases in ranges from 25% to just over 40% for the AS, CSI and DS models, except the SCSI model, which remains similar to the training model. The error variances of the training data compared to the test data suggest a good response for all models, except for the CSI, which has a larger deviation for data that contain new observations which the model was not able to learn.

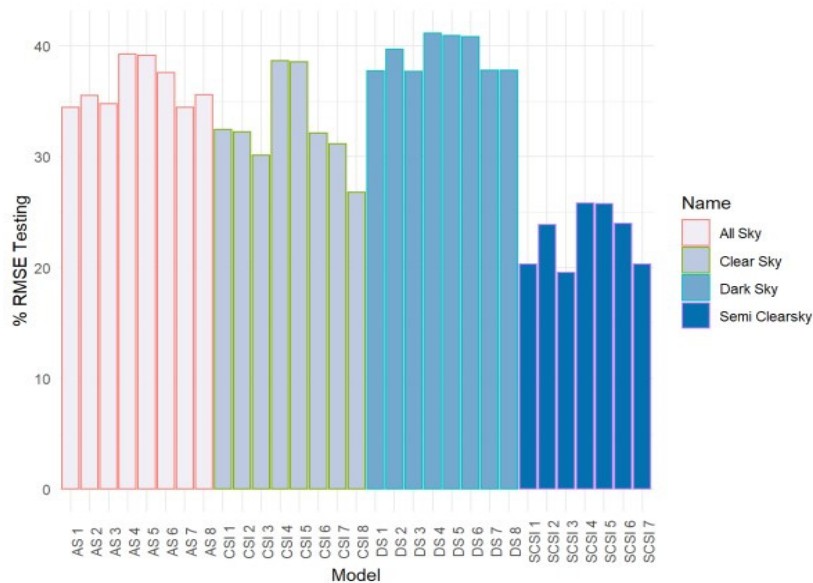

**Figure 18.** RMSE for testing models.

Figure 19 details each of the testing models generated, with their MBE bias errors. This indicator warns of the overestimation or underestimation of the model concerning the dependent variable. The training MBE errors can be considered negligible. It can be seen in image 19 that the test errors mostly underestimate the model, ranging from about 0% to 3%, except for the CSI models, which show a significant increase in overestimation, varying from 8% to 14%.

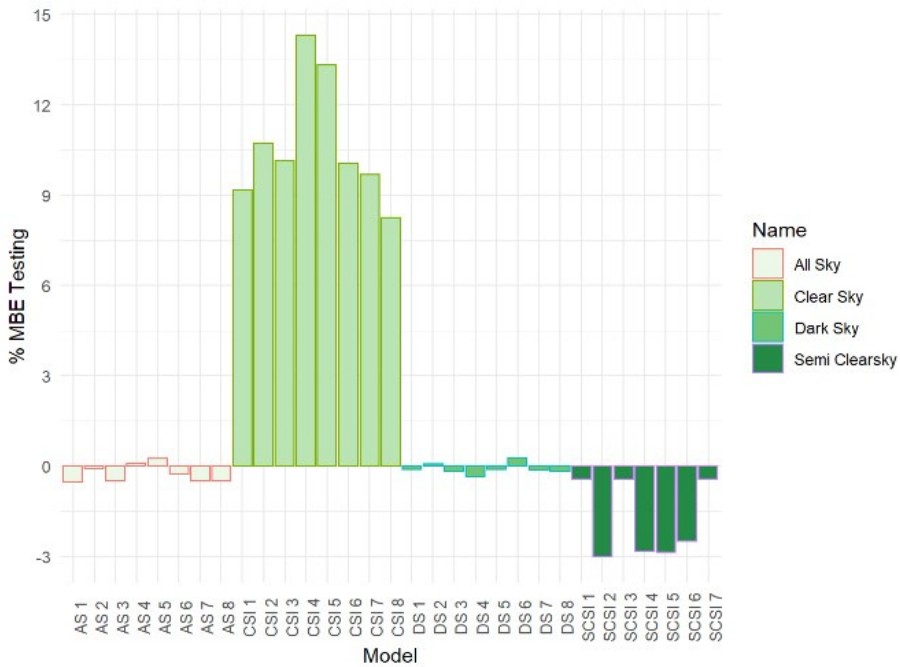

**Figure 19.** MBE for testing models.

### 3.4. Selection and Evaluation of the Optimal Cloudiness Model

The AS 7 model was selected as suitable, because it explains all sky conditions. It uses five predictor variables, and although it does not have the lowest test RMSE; the difference with the training errors is below 2%. When applying the AS 7 model, a new fit dataset for the GHI data is generated, which replaces the original satellite data (site-adaptation). The final dataset generated was compared to the surface measurements, and the coefficient of determination $R^2$ increased from 0.607 to 0.876 for the hourly GHI profile. Figure 20 shows the dispersion reduction after site-adaptation process.

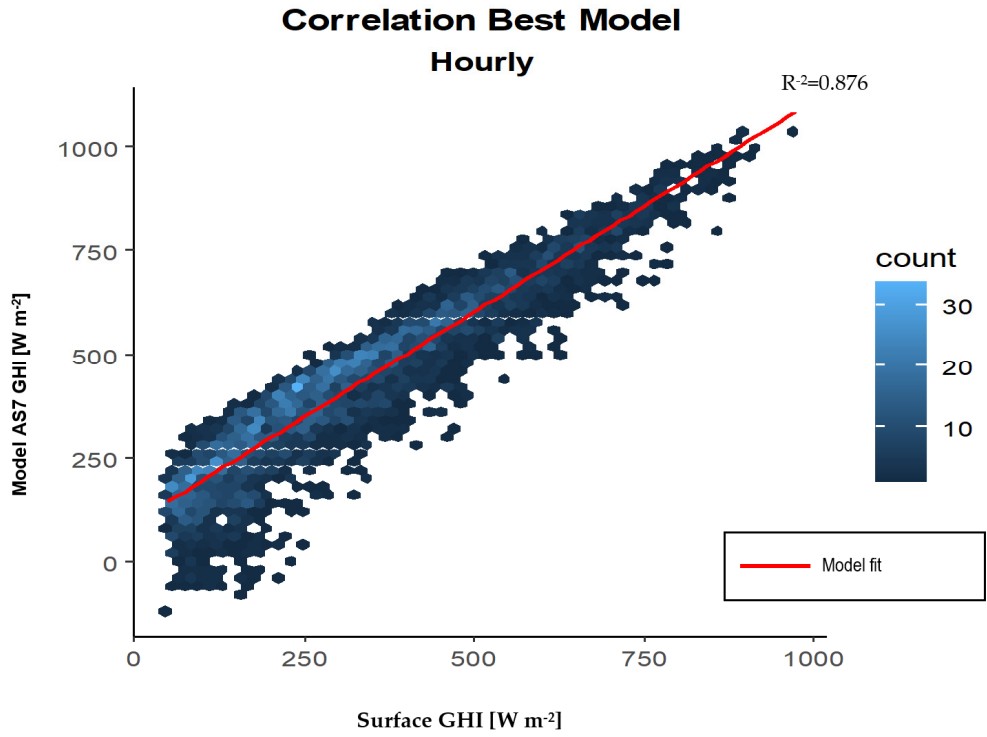

**Figure 20.** Correlation between Surface GHI and best model selected for all sky.

## 4. Discussion

The WMO methodology applied for the debugging of the surface and satellite data series was necessary for quality assurance. The process of verification and subsequent filtering of GHI data (satellite and surface) represents the cornerstone for site validation and adaptation. Removing night hours, identifying extremely rare, physically possible values for filtering and filling gaps with interpolation methods allow reliable handling of GHI data sets. Missing this procedure may cause modeling biases. The results of the verification and debugging stage show the small amount of outliers and the absence of influential points.

High variability of solar radiation in short temporal resolution profiles makes it difficult to use satellite estimates for simulation purposes. The literature suggests that before using solar radiation information from remote sensing, it is necessary to validate it, especially in high-cloud-cover areas. The satellite GHI validation methods are generally performed by non-parametric statistical indicators. However, graphical analysis enables the comparison of the hourly and monthly profiles defined after validation, showing greater variability for the hourly profiles. In this process, clusters could be identified for the application of site adaptation techniques such as quantile mapping.

The results of parametric tests between hourly and monthly profiles allow quantifying the main differences in both profiles. The validation allows to identify by means of statistical parameters whether the GHI profile coming from satellites can be employed reliably in energy production simulation. In that sense, the parametric graph analysis shows insignificant differences in the monthly profiles. The tests performed reflect the fulfillment of all parameters except for normality, which is consistent due to its temporal variability and atmospheric attenuation. This result leads to performing a non-parametric analysis.

The dispersion statistical indicators RMSE and MBE applied in the non-parametric analysis show the overestimation of the satellite GHI. The KS-test proves that the hourly profile could not be reliably used for energy production simulation purposes. These results are in agreement with the findings in [25] and also in parametric graphic analysis. The identification of the profile that does not comply with the validation allows initiating the process of site-adaptation.

The 31 models generated for site-adaptation employ techniques ranging from heuristic to the application of statistical learning using machine learning algorithms. The input variables use solar geometry variables, GHI surface measurements and satellite-collected parameters. The models developed could be used for any type of sky. The cloudiness hypothesis based on the clearness index shows that for clear sky models there is a higher adjusted $R^2$ and lower dispersion as expected. Site-adaptation through the AS7 model significantly improves the adjusted $R^2$.

The results in [18] show a better performance of the GHI from CAMS for clear-sky models, with an MBE < 1% and RMSE < 5%. In this study, the clear-sky models CSI presented an MBE < 15%; however, the all-sky models AS and cloudy-sky models DS presented an MBE < 1%, whereas the lower RMSE was found in the SCSI models with an RMSE < 26%. Although the research findings [16] are given on daily and weekly profiles, the models based on machine learning are notably higher than those based on linear regressions and QM. R2 values of 0.91 and 0.88 are obtained for daily and weekly profiles, respectively. In agreement with these results, this paper presents improved performance with the machine learning algorithms Best Subset Selection and Forward Stepwise Selection, raising the R2 of the satellite estimates of NREL to 0.876 for hourly profiles. Several studies have shown the improvement in satellite data using different site-adaptation methodologies. However, climatic zones with annual cloudiness above 80% have not been evaluated. In that sense, the present work highlights the complexity of creating a generic model that responds to high cloud cover areas.

Future research could add surface measurements of direct and diffuse irradiance to the model to increase its predictive capability. Additionally, the site-adaptation methodology used could generate models for the assessment of photosynthetically active radiation (PAR), which is necessary to find suitable locations of biomass potential. Further work could also focus on developing site-adaptation techniques based on genetic algorithms and artificial intelligence. As well as testing the behavior of the model in different geographies with high cloud cover, one could add more variables to analyze the model performance. Finally, this study may be able to generate new physical models based on regional adaptation using probabilistic methods.

## 5. Conclusions

An innovative site-adaptation technique capable of improving the GHI from the NREL database for high-cloud-cover areas was developed. Surface measurements, solar geometry and data from the NREL database were used. The verification, parametric and non-parametric validation processes identified the satellite GHI hourly profile as invalid because of its high temporal variability. With the identified profile and the best correlation input variables, the profile was segmented according to the clearness index Kt, which considers different sky conditions. The use of statistical learning has indicated the AS 7 model as the most suitable, by evaluating it with the lowest RMSE and MBE test and training statistics, considering the highest adjusted $R^2$.

The debugging process allowed the surface and satellite GHI data series to be compared, eliminating outliers. This effect is evidenced in the results in Section 3.2 with the absence of influential points.

- The validation of the satellite GHI from the NREL database considered a parametric and non-parametric analysis. The parametric linearity analysis showed a linear trend for the hourly and monthly GHI profiles considered. The hourly profile had a lower $R^2$ compared to the $R^2$ of the monthly profile.
- A comparison of medians between the surface GHI data series versus the satellite GHI data series showed an overestimation of the solar resource in the satellite GHI.
- The parametric normality tests showed that the hourly and monthly profiles evaluated did not satisfy the defined parameters; therefore, a non-parametric validation analysis was necessary.
- The parametric variance of the residuals test revealed that only the monthly profile complied with homoscedasticity.
- Parametric tests for influential points revealed in both the hourly and monthly GHI profiles the absence of outliers or anomalous values that could have an impact on the prediction model.
- Non-parametric and dispersion statistics analysis identified the hourly profile as inapplicable for simulation purposes. From the hourly profile, the site-adaptation process was performed.

In locations with high cloud cover, the use of satellite images for solar resource assessment represents a challenge to obtain fine-resolution profiles. In response to this, a site-adaptation methodology was applied. Through linear, non-linear and statistical learning regressions, 31 empirical models were generated for different sky types of the Ecuadorian coast. These models were based on the calculation of solar geometric characteristics and information from remote sensing variables available in the NREL database.

The application of the models represents a relevant contribution, not only for areas of high cloud cover but also for more favorable conditions for remote sensing.

Models that use machine learning applying the "Forward Stepwise Selection" algorithm denote a higher adjusted $R^2$ than the other models, but not necessarily a lower RMSE test error.

The AS 7 all-sky model was selected due to its higher adjusted $R^2$ of 0.697 and its testing RMSE of 34.46%, a gap of 1.78% with respect to training RMSE errors. The final model was evaluated against surface measurements, increasing the adjusted $R^2$ from 0.607 to 0.876, suggesting a better fit for the initially evaluated hourly GHI satellite profile.

**Author Contributions:** Concept, M.M.-S. and F.P.-L.; methodology and software, M.M.-S. and F.P.-L.; formal analysis, F.P.-L. and F.T.; research, all the authors; writing—original draft preparation, M.M.-S. and F.P.-L.; writing—review and editing, F.P.-L. and F.T. All authors have read and agreed to the published version of the manuscript.

**Funding:** This research received no external funding.

**Data Availability Statement:** The surface GHI data presented in this study are available on request from the corresponding author due to [privacy restrictions]. The satellite GHI data are available [https://nsrdb.nrel.gov/data-viewer, 13 September 2022].

**Acknowledgments:** Special gratitude goes to the Carolina Foundation, the Universidad Estatal de Milagro and the Universidad de Córdoba, institutions that have co-funded the author's doctoral studies.

**Conflicts of Interest:** The authors declare no conflict of interest.

## Abbreviations

| | |
|---|---|
| **AS** | All Sky |
| **CAMS** | Copernicus Atmosphere Monitoring Service |
| **CIM** | Cloud Index Method |
| **COP26** | Conference of the Parties |
| **CSI** | Clear Sky |
| **DNI** | Direct Normal Irradiance |
| **DS** | Cloudy Sky |
| **FARMS** | Fast All-sky Radiation Model for Solar applications |
| **GHI** | Global Horizontal Irradiation [W m$^2$] |
| **GOES** | Geostationary Operational Environmental Satellites |
| **GLMNET** | Elastic Net regression |
| **INHAMI** | National Institute of Meteorology and Hydrology |
| **KS-test** | Kolmogorov–Smirnov test |
| **Kt** | Clearness Index |
| **MARS** | Multivariate Adaptive Regression Splines |
| **MBE** | Mean Bias Error |
| **MOS** | Model Output Statistics |
| **NREL** | National Renewable Energy Laboratory |
| **NSRDB** | National Solar Radiation Database |
| **ODR** | Orthogonal Distance Regression |
| **PSM** | Physical Solar Model |
| **RF** | Random Forests |
| **RMSE** | Root Mean Square Error |
| **SCSI** | Semi-Clear Sky |
| **SDE** | Stochastic Differential Equation |
| **SDR** | Symphonie Data Retriever |
| **SVR** | Support Vector Regression |
| **UNEMI** | State University of Milagro |
| **WMO** | World Meteorological Organization |
| **WRF** | Weather and Research Forecasting |
| **XGBoost** | Extreme Gradient Boosting Machines |

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
