# Peer review of "An Empirical Correction Model for Remote Sensing Data of Global Horizontal Irradiance in High-Cloudiness-Index Locations"

_remotesensing, doi:10.3390/rs14215496_

Round 1
Reviewer 1 Report
Dear authors,
I would like to congratulate you on a really interesting work that provides good results. In my opinion, the redaction style and structure makes the paper easy to read and understand. Next, I will try to present you some opinions, doubts and suggestions.
General comments.
In my opinion, the "Introduction" seems to me too long (5 pages) and, perhaps, could be restructured as an “Introduction + "previous works/state of the art".
On the other hand, structurally four levels of sections/epigraphs seems excessive to me. In any case, each section/subsection should introduce a text and not directly a new sublevel.
The nomenclature used in the text is adequate for solar energy papers but could be improved by following the recommendations suggested in "Units and symbols in Solar Energy", Solar Energy 73, III-V (https://doi.org/10.1016/S0038-092X(02)00081-6). Why don't you use subscripts/superscripts for variable declaration/assignment? In text citations, in models description Tables… improves the visual interpretation.
You must choose a unique symbol as decimal separator for the whole document.
About solar radiation language, the scientific community (in general) uses the term "clearness index" to refer to the GHI/I0 ratio ...I have never seen "clarity index" or "transparency index" in this context. Similarly, "solar elevation angle" is a better term (and less confusing) than "solar height".
When you write “turbidity index”, are you referring to the “Linke turbidity index”?
The GHI provided by NREL database is an estimation from satellite images, but never a measurement. Please, review the whole document including Tables and Figures.
Introduction.
In Table 1, you must introduce sign criteria for geographical coordinates or add North-South East-West information.
Materials and methods comments.
About the databases used, I miss a more detailed description of the data used: extent of both databases, sampling/recording frequency (ground database); spatial/time resolution, nadir coordinates, relative position of the radiometric station with respect to the pixel representing it (satellite estimations),...
Starting this section (232-234) you write “…The first stage…The goal was to remove outliers by comparing surface and satellite GHI variables…” Do you identify the outliers in the residuals and subsequently remove these records in both databases? Or do you identify the outliers in each database and remove the records corresponding to these positions in both databases?
In #269 “2.2.1 Data set normalization” you do not describe any normalization process…you are describing a filtering process according to BSRN quality analysis proposal.
In line #291 “For the monthly profile, the data set was grouped according to time t, where ? ∈ [?0, ?]∁[0, 23]” in line brackets to introduce a temporal interval could be confused taking account than in line brackets also used for bibliographic references citations. In any case, are you introducing a monthly profile as a set of daily values?
In “Table 3. Model predictors considered”, the variables declaration as xi (i=1,…8) and y is an author’s choice, clear and correct but could be improved by using the usual symbols/nomenclature in solar energy texts: x1 = GHI, x2 = α, x3 = ϒ, x4 = kt… most of them previously defined. Below, Table 5, the proposed models will be more clearly understood.
From a mathematical point of view, the proposal and fitting of multivariate models do not necessarily have to take into account the quantities/dimensions/units of the explanatory variables. From a physical point of view, the proposed models should be coherent with respect to the quantities/dimensions/units... it is more "elegant" in any case. Why did you not choose dimensionless variables for model fitting?
You present (line #368 et seq.) a sky classification by clearness index levels: “…Kt > 0.7 were proposed for the first CSI section... 0.5< Kt < 0.7 were set for SCSI… values of Kt < 0.5 were used in the DS model…”. Is this your personal/empirical choice or it’s based in previous unreferenced works? Justify it please.
In #376-377 lines you write “The training data were randomly generated and corresponded to 80% of the total stretch, while the test 377 data corresponded to 20% of the total stretch, also obtained randomly”. I understand that 20% of the data selected for model validation correspond to the rest of data, that is, they are a complementary set to the 80% randomly chosen for training and are not randomly chosen again; this sentence is confusing in my opinion.
Results
In #411 “Normalization” I must insist that you are presenting the result of applying a quality test.
After applying the BSRN Test on GHI ground data, you found a 6% of missing/wrong data. We’ve got 8760 hourly records per year, 4380 records corresponding to daylight period and 4350 high quality data so only 30 values have been identified as missing/missing data… I don’t understand your calculations or maybe 30 values were not interpolated (???)
In any case, you should clarify how many values did not pass the BSRN Test, which interpolation criteria were used (how many consecutive gaps, full days or not...) and which interpolation method was applied (linear, splines,...). The same information for NREL database.
In line #425-426 the same comment above about brackets for the same expression "For the monthly profile, the dataset was aggregated as a function of time t, where ? ∈ [?0, ?]∁[0, 23]". And same question... When you write "monthly profile" do you mean a set of daily values? In this case, figure captions as “Figure 6. Scatterplot of monthly surface vs. satellite GHI profiles” are confusing as the figure shows more than 12 points (probably 365 daily values) and it is not possible to identify any month.
If “Figure 6.” shows daily values, does it shows “average daily values”? In my opinion, “daily solar radiation” as the sum of hourly values is more adequate but, in any case, if you choose average values for solar radiation you must explain if the average is calculated over 24h or daylight period (in Ecuador daylight is ~12h all yearlong, is true). You should consider these comments for the whole text.
On “Figure 7.” are you show residuals values vs. predicted values? What do you mean by “predictions” as x axis label?
In Table 5, in my opinion, the statistical results of the training data are not necessary. If this information is removed, the table will look clearer, the equations describing the models will look better.... Regarding these equations, without the use of subscripts/superscripts they are confusing.
In “Figure 17: Best subset selection algorithm of number of variables for all Kt sections” what represent the colors scale?
For Figure 18 and Figure 19, as my previous comments about Table 5, information related to training data is not relevant.
References
Bibliographic references should have a unified format. Some of them are confusing, incomplete, erroneous and/or impossible to consult... e.g. 3, 5, 12, 19, 21, 26, 33, 34
Author Response
The authors are grateful for all the comments and recommendations made in this work. Your contribution will definitely give more relevance to the manuscript.
General comments.
In my opinion, the "Introduction" seems to me too long (5 pages) and, perhaps, could be restructured as an “Introduction + "previous works/state of the art".
We accept the suggestion and proceed with the change (94-217) by splitting the state of the art.
On the other hand, structurally four levels of sections/epigraphs seems excessive to me. In any case, each section/subsection should introduce a text and not directly a new sublevel.
It is an excellent suggestion to avoid confusion in the reader. We remove the sublevels (338, 366, 380, 448, 524, 728) and instead we describe them at the beginning of each paragraph.
The nomenclature used in the text is adequate for solar energy papers but could be improved by following the recommendations suggested in "Units and symbols in Solar Energy", Solar Energy 73, III-V (https://doi.org/10.1016/S0038-092X(02)00081-6). Why don't you use subscripts/superscripts for variable declaration/assignment? In text citations, in models description Tables… improves the visual interpretation.
The suggestion is accepted and the nomenclature is improved in the whole text.
You must choose a unique symbol as decimal separator for the whole document.
We appreciate the observation. The suggestion is accepted and the dot is used as decimal separator in the whole text.
About solar radiation language, the scientific community (in general) uses the term "clearness index" to refer to the GHI/I0 ratio ...I have never seen "clarity index" or "transparency index" in this context. Similarly, "solar elevation angle" is a better term (and less confusing) than "solar height".
We admire your mastery of the subject. We have replaced the suggested terms in the whole document.
When you write “turbidity index”, are you referring to the “Linke turbidity index”?
We refer to the Linke turbidity index. We have made the change and added its reference (112).
The GHI provided by NREL database is an estimation from satellite images, but never a measurement. Please, review the whole document including Tables and Figures.
We are in full agreement, thank you very much. The changes have been made in the whole document including tables and figures.
Introduction.
In Table 1, you must introduce sign criteria for geographical coordinates or add North-South East-West information.
We appreciate your observation, the sign criteria has been defined (234, 235) and maintained in the whole document.
Materials and methods comments.
About the databases used, I miss a more detailed description of the data used: extent of both databases, sampling/recording frequency (ground database); spatial/time resolution, nadir coordinates, relative position of the radiometric station with respect to the pixel representing it (satellite estimations),...
Thank you for your comment. A more detailed description of the data used: extent of both databases, sampling/recording frequency (surface database); spatial/temporal resolution, zenith angle coordinates, relative position of the radiometric station with respect to the pixel, was added in the sections (318, 327, 331, 332, 333).
Starting this section (232-234) you write “…The first stage…The goal was to remove outliers by comparing surface and satellite GHI variables…” Do you identify the outliers in the residuals and subsequently remove these records in both databases? Or do you identify the outliers in each database and remove the records corresponding to these positions in both databases?
We identified the outliers in each database and eliminated the records corresponding to these positions in both database. This explanatory text was added in ( 294, 295).
In #269 “2.2.1 Data set normalization” you do not describe any normalization process…you are describing a filtering process according to BSRN quality analysis proposal.
Totally agree. The proper term is "verification" and subsequent debugging. This clarification has been applied to the whole document. Thanks a lot.
In line #291 “For the monthly profile, the data set was grouped according to time t, where ? aily values [?0, ?]∁[0, 23]” in line brackets to introduce a temporal interval could be confused taking account than in line brackets also used for bibliographic references citations. In any case, are you introducing a monthly profile as a set of daily values?
It is true that the use of brackets is confusing. We introduce a monthly profile as a set of hourly values. That is, a monthly average hourly profile. Where each month has 24 values. This clarification was performed in (363, 527, 528).
In “Table 3. Model predictors considered”, the variables declaration as xi (i=1,…8) and y is an author’s choice, clear and correct but could be improved by using the usual symbols/nomenclature in solar energy texts: x1 = GHI, x2 = α, x3 = ϒ, x4 = kt… most of them previously defined. Below, Table 5, the proposed models will be more clearly understood.
We appreciate the suggestion. The nomenclature for variable declaration has been modified (Table 3). And consequently the equations in Table 5.
From a mathematical point of view, the proposal and fitting of multivariate models do not necessarily have to take into account the quantities/dimensions/units of the explanatory variables. From a physical point of view, the proposed models should be coherent with respect to the quantities/dimensions/units... it is more "elegant" in any case. Why did you not choose dimensionless variables for model fitting?
Very useful observation. We believe that for a better understanding of the site-adaptation effect, it is easier to analyze the biases in terms of quantities and units.
You present (line #368 et seq.) a sky classification by clearness index levels: “…Kt > 0.7 were proposed for the first CSI section... 0.5< Kt < 0.7 were set for SCSI… values of Kt < 0.5 were used in the DS model…”. Is this your personal/empirical choice or it’s based in previous unreferenced works? Justify it please.
The choice of splitting Kt : "...Kt > 0.7 were proposed for the first CSI section.... 0.5< Kt < 0.7 were set for SCSI... values of Kt < 0.5 were used in DS mode..." obeys an empirical criterion and the distribution of the data set according to Kt, however a paper using other Kt levels is cited. The justification is found in (450, 457).
In #376-377 lines you write “The training data were randomly generated and corresponded to 80% of the total stretch, while the test 377 data corresponded to 20% of the total stretch, also obtained randomly”. I understand that 20% of the data selected for model validation correspond to the rest of data, that is, they are a complementary set to the 80% randomly chosen for training and are not randomly chosen again; this sentence is confusing in my opinion.
Totally agree, the clarification has been made in (459, 461).
Results
In #411 “Normalization” I must insist that you are presenting the result of applying a quality test.
Agreed, the mistake was corrected previously.
After applying the BSRN Test on GHI ground data, you found a 6% of missing/wrong data. We’ve got 8760 hourly records per year, 4380 records corresponding to daylight period and 4350 high quality data so only 30 values have been identified as missing/missing data… I don’t understand your calculations or maybe 30 values were not interpolated (???)
Good point. 30 values were not interpolated, this clarification has been made in (513, 514).
In any case, you should clarify how many values did not pass the BSRN Test, which interpolation criteria were used (how many consecutive gaps, full days or not...) and which interpolation method was applied (linear, splines,...). The same information for NREL database.
The comment is accepted and the suggested information is added (512, 522).
In line #425-426 the same comment above about brackets for the same expression "For the monthly profile, the dataset was aggregated as a function of time t, where ? ∈ [?0, ?]∁[0, 23]". And same question... When you write "monthly profile" do you mean a set of daily values? In this case, figure captions as “Figure 6. Scatterplot of monthly surface vs. satellite GHI profiles” are confusing as the figure shows more than 12 points (probably 365 daily values) and it is not possible to identify any month.
The change has been previously performed. For simulation purposes, it is common to obtain monthly average hourly profiles. In this case 144 values by removing night hours.
If “Figure 6.” shows daily values, does it shows “average daily values”? In my opinion, “daily solar radiation” as the sum of hourly values is more adequate but, in any case, if you choose average values for solar radiation you must explain if the average is calculated over 24h or daylight period (in Ecuador daylight is ~12h all yearlong, is true). You should consider these comments for the whole text.
As in the previous response, the monthly profile has been conformed as a set of hourly averages.
On “Figure 7.” are you show residuals values vs. predicted values? What do you mean by “predictions” as x axis label?
We appreciate the observation. The correct is predicted. Corrections are made on the x-axis of figure 7.
In Table 5, in my opinion, the statistical results of the training data are not necessary. If this information is removed, the table will look clearer, the equations describing the models will look better.... Regarding these equations, without the use of subscripts/superscripts they are confusing.
Although comparison of the training errors with the test errors could reveal interesting differences, we are in full agreement that removing the training RMSE and MBE columns, the table improves the visualization of the equations. The changes have been made in Table 5 and the subscripts and superscripts have been placed.
In “Figure 17: Best subset selection algorithm of number of variables for all Kt sections” what represent the colors scale?
The colors follow only the aesthetics of the graphic software but do not represent any relation with the variables.
For Figure 18 and Figure 19, as my previous comments about Table 5, information related to training data is not relevant.
According to the above criteria, the training RMSE and MBE plots have been removed.
References
Bibliographic references should have a unified format. Some of them are confusing, incomplete, erroneous and/or impossible to consult... e.g. 3, 5, 12, 19, 21, 26, 33, 34
According to your criteria, references 3, 5, 12, 19, 21, 26, 33, 34 have been rectified.
Reviewer 2 Report
A further explanation of the statistical learning used in index 7 and 8 is necessary.
Compared to the kind explanation of research process and the results, the discussion of the research results and the suggestion of new ideas are insufficient.
It is hoped that an optimal model that can have originality compared to existing studies can be presented.
Figure 2. 80% for model training and 30% for model testing → 20% for model testing
Figure 3 : It would be better to show the numbers as well
Author Response
The authors greatly appreciate the time spent in reviewing the manuscript and are confident that the observations and comments provided will give it greater relevance
A further explanation of the statistical learning used in index 7 and 8 is necessary.
We fully agree with the recommendation. A more detailed explanation of the statistical learning algorithms "Best Subset Selection" and "Forward Stepwise Selection" is given in lines (478-496).
Compared to the kind explanation of research process and the results, the discussion of the research results and the suggestion of new ideas are insufficient.
We appreciate your accurate criteria. Further discussion of the results and new ideas for future work are given in sections (840-864, 895-900).
It is hoped that an optimal model that can have originality compared to existing studies can be presented.
According to his comment, the comparison of the model with similar works has been carried out, highlighting the novelty of our research in sections (878-891)
Figure 2. 80% for model training and 30% for model testing → 20% for model testing
We appreciate your feedback. The mistake has been fixed in figure 2.
Figure 3 : It would be better to show the numbers as well
According to your suggestion, correlation values are added to Figure 3.
Reviewer 3 Report
An innovative site-adaptation technique capable of improve the GHI from the NREL database for high cloud cover areas was developed. Surface measurements, solar geometry and data from the NREL database were used. The normalization, parametric and non-parametric validation processes identified the satellite GHI hourly profile as invalid. With the identified profile and the best correlation input variables, the profile was segmented according to the transparency index Kt, which considers different sky conditions. The use of statistical learning has indicated the AS7 model as the most suitable, by evaluating it with the lowest RMSE and MBE test and training statistics, considering the highest adjusted R2. My comments and suggestions are as follows.
Line 13, it is suggested to rewrite the abstract and show main contents in this study.
Line 217, more and clear introduction about the main objectives are needed in this paragraph, and let the readers know them clearly at the beginning. Such as site-adapted satellite GHI models from the surface measurement for all sky conditions were developed.
Lines 311 and 317, please check equations 3 and 4.
For model validations, please compare other empirical models for hourly and monthly RMSE and MAE values, for a better understanding of the model performance.
Line750,Non-parametric and dispersion statistics analysis identified the hourly profile as inapplicable for simulation purposes, in other words, does it mean that the models do not perform very well for hourly GHI compared to monthly GHI? It is suggested to be explained in the text.
Author Response
The authors are especially grateful to the reviewer for the comments and suggestions made on the manuscript, which will certainly improve its quality.
Line 13, it is suggested to rewrite the abstract and show main contents in this study.
The suggestion is accepted and the abstract is rewritten according to the main contents of the manuscript in section (13-35).
Line 217, more and clear introduction about the main objectives are needed in this paragraph, and let the readers know them clearly at the beginning. Such as site-adapted satellite GHI models from the surface measurement for all sky conditions were developed.
Section (276-293) explains in more detail the objectives of the present study for the development of a site adaptation model for any sky condition.
Lines 311 and 317, please check equations 3 and 4.
Equations (3) and (4) have been realized, the redundant percentage symbol has been removed.
For model validations, please compare other empirical models for hourly and monthly RMSE and MAE values, for a better understanding of the model performance.
The suggestion is accepted and the comparison with similar studies is made in lines (870-875, 883-896).
Line750,Non-parametric and dispersion statistics analysis identified the hourly profile as inapplicable for simulation purposes, in other words, does it mean that the models do not perform very well for hourly GHI compared to monthly GHI? It is suggested to be explained in the text.
The site-adaptation process begins by identifying the profile that does not comply with the validation, in this case the hourly profile. This condition was defined in sections (27-31, 288-289, 407-408).
Round 2
Reviewer 2 Report
1. Please explain why the scsi 8 model is missing.
2. There are many other machine learning algorithms, but I would like to know why "best sebset selection" and "forward stepwise selection" were used in preference in this paper.
Author Response
Please explain why the scsi 8 model is missing.
We are very grateful for the observation. The equation and statistical indicators of the SCSI 8 model are the same as the SCSI 7 model. The SCSI 8 model has been placed on Table 5 and the explanation has been detailed on lines (642-644).
2. There are many other machine learning algorithms, but I would like to know why "best subset selection" and "forward stepwise selection" were used in preference in this paper.
Thank you for your inquiry. Unlike common linear models that use least squares regression, both algorithms have been used due to the following criteria: a) number of predictors found, b) cross-validation, which finds the optimal combination of predictors and avoids model overfitting, and c) computing efficiency. This clarification has been added in (398-401).